# Downregulation of rhodopsin is an effective therapeutic strategy in ameliorating peripherin-2-associated inherited retinal disorders

Christian T. Rutan Woods [1], Mustafa S. Makia[1], Tylor R. Lewis [2], Ryan Crane[1], Stephanie Zeibak [1], Paul Yu [1], Mashal Kakakhel [1], Carson M. Castillo[2], Vadim Y. Arshavsky [2], Muna I. Naash [1] ✉ & Muayyad R. Al-Ubaidi [1] ✉

Given the absence of approved treatments for pathogenic variants in Peripherin-2 (*PRPH2*), it is imperative to identify a universally effective therapeutic target for *PRPH2* pathogenic variants. To test the hypothesis that formation of the elongated discs in presence of PRPH2 pathogenic variants is due to the presence of the full complement of rhodopsin in absence of the required amounts of functional PRPH2. Here we demonstrate the therapeutic potential of reducing rhodopsin levels in ameliorating disease phenotype in knockin models for p.Lys154del (c.458-460del) and p.Tyr141Cys (c.422 A > G) in *PRPH2*. Reducing rhodopsin levels improves physiological function, mitigates the severity of disc abnormalities, and decreases retinal gliosis. Additionally, intravitreal injections of a rhodopsin-specific antisense oligonucleotide successfully enhance the physiological function of photoreceptors and improves the ultrastructure of discs in mutant mice. Presented findings shows that reducing rhodopsin levels is an effective therapeutic strategy for the treatment of inherited retinal degeneration associated with *PRPH2* pathogenic variants.

Inherited retinal diseases (IRDs) encompass a wide range of heterogeneous disorders (https://web.sph.uth.edu/RetNet/disease.htm) resulting in the degeneration of photoreceptors and progressive visual impairment worldwide[1–3]. Advancements in genetic engineering have shed light on their etiology, leading to the development of the first FDA-approved IRD gene therapy and various clinical trials[1,4].

Of the many genes associated with IRDs, mutations in *Peripherin-2* (*PRPH2*) make up some of the most prevalent disease-causing pathogenic variants with over 200 already identified[5,6]. PRPH2 is a photoreceptor-specific tetraspanin located in the photoreceptor outer segment (OS) disc rim region and forms homo-tetramers as well as hetero-tetramers with its homolog, rod outer segment membrane

protein 1 (ROM1)[7,8]. These tetramers subsequently assemble into octamers and higher order oligomers, which play a crucial role in OS disc rim formation[9]. Minimum threshold PRPH2 protein levels (~80% of wild type) are known to be essential for OS disc morphogenesis and maintenance[10]. Therefore, mice heterozygous for the *Prph2* null allele (*Prph2*[+/-]) exhibit highly abnormal OSs, while homozygotes (*Prph2*[-/-]) fail to develop OSs[11,12].

To date, there are no approved treatments or ongoing clinical trials specifically for *PRPH2* associated diseases[6]. While many studies have provided insights into the correlation between *PRPH2* genotypes and the resulting phenotypes in animal models[13–18], establishing these associations in human patients is challenging. This difficulty arises from the

[1]Department of Biomedical Engineering, University of Houston, Houston, TX 77204, USA. [2]Department of Ophthalmology, Duke University Medical Center, Durham, NC 27710, USA. ✉e-mail: mnaash@central.uh.edu; malubaid@central.uh.edu

substantial degree of interfamilial and intrafamilial phenotypic variability observed among individuals carrying identical *PRPH2* pathogenic variants and, the complexity is further compounded by the occurrence of digenic inheritance involving its binding partner ROM1[19–24]. *Prph2* mutations can exert their effects through loss-of-function, dominant-negative, and/or gain-of-function mechanisms[13,14,16], making traditional gene supplementation therapies inapplicable for all pathogenic variants. Supplementation alone cannot effectively treat dominant-negative and gain-of-function pathogenic variants[13], while combining gene knockdown with supplementation could be a potential solution. However, the large number of unique pathogenic variants and their low occurrence renders this approach economically unfeasible[25]. These complex pathogenic mechanisms along with the vast array of different disease-causing pathogenic variants makes the identification of a ubiquitous target for the treatment of *PRPH2*-associated diseases highly desirable.

The ratio of PRPH2 to the rod-specific protein rhodopsin (RHO) plays a crucial role in morphogenesis of rod OS discs[26,27]. RHO is a light-sensitive protein located in the disc lamellae and is the most abundant retinal protein[28,29]. It is highly regulated due to its essential role in phototransduction and visual function[27,30–33]. Murine models with an increased RHO/PRPH2 ratio exhibited formation of discs with significantly larger diameters than wild type, accelerated retinal degeneration, and decreased physiological function[26,27,32]. In contrast, models with a decreased ratio display properly oriented rod outer segments (ROSs) with decreased diameters[27,34,35] but possess reduced sensitivity and faster flash-response kinetics before degenerating over a slow time course[34–36]. Through the utilization of a mouse model that overexpresses PRPH2, we successfully alleviated the stress associated with abnormally high RHO:PRPH2 ratio, which emphasized the significance of maintaining this ratio for proper photoreceptor structure and function[27]. It is imperative to acknowledge that any therapeutic interventions targeting PRPH2-affected photoreceptors must focus on generating sufficient protein levels to maintain the RHO:PRPH2 ratio as close as it is in wild type retinas. These findings also shed light on the delicate nature of rod photoreceptors, indicating that their heightened sensitivity to excessive opsin levels. Pathogenic variants in *PRPH2* are known to decrease the levels of functional protein contributing to the formation of the elongated discs[37]. Therefore, we hypothesized that an increased RHO to functional PRPH2 ratio contributes to disc elongation and disease progression of *PRPH2*-associated disorders.

In this study, we demonstrate that modifying the RHO/PRPH2 ratio in favor of PRPH2 positively impacts the disease phenotype in previously characterized knockin mouse models expressing the patient pathogenic variants c.461_463del, p.Lys154del[19] (herein called K153Δ for the knockin mouse model[13]) and c.422 A > G, p.Tyr141Cys[22,23] (herein called Y141C for the knockin mouse model[14]) in *PRPH2*. Proof-of-concept studies presented here revealed that reducing one allele of *Rho* in the heterozygous models improves photoreceptor ultrastructure and physiological function. Furthermore, in order to demonstrate the translational applicability of this strategy, we employed a previously characterized antisense oligonucleotide (ASO) *mRho* ASO1 shown to effectively reduce *Rho* transcript levels following intravitreal administration[38]. ASO-mediated reduction of RHO levels in mice heterozygous for the *Prph2* Y141C knockin mutation resulted in improved retinal function, OS ultrastructure, and delayed photoreceptor degeneration. These findings suggest that reducing RHO levels is a potential therapeutic strategy to ameliorate the disease phenotype in patients with *PRPH2*-associated inherited retinal disorders.

## Results

### Reducing RHO levels in *Prph2* knockin mouse models expressing mutant PRPH2 leads to functional improvements

In *Rho*[+/-] mice, retinal development proceeds normally with proper lamination but gradually undergoes degeneration over time[34].

The reduction of RHO manifests specific effects on the ultrastructure of ROS, including an elliptical disk shape and decreased surface area[35]. To determine the impact of reducing RHO levels in ROS discs on functionality of the retinas from mice expressing mutant forms of PRPH2, comprehensive electroretinography measurements (ERG) were conducted on *Prph2*[K153Δ/+] and *Prph2*[Y141C/+] mice that were also hemizygous for *Rho* (*Rho*[+/-]). These *Prph2* heterozygous mice were chosen for their clinical relevance, as *PRPH2*-associated retinitis pigmentosa typically exhibits autosomal dominant inheritance[16,39]. ERGs were performed at various postnatal (P) stages (P17, P30, and P90) to evaluate differences in scotopic and photopic maximum amplitudes during photoreceptor maturation and disease progression (Fig. 1A–F).

Eliminating one allele of *Rho* in *Prph2*[Y141C/+] mice led to a significant improvement in mean maximum amplitudes of scotopic a- and b-waves as early as P17 compared to *Prph2*[Y141C/+] controls (~53% and ~43% increases respectively, Fig. 1C). Additionally, there was an increase in the photopic (~29%) b-wave mean maximum amplitude, although this increase did not reach statistical significance (Fig. 1D). These improvements persisted throughout the study, as *Prph2*[Y141C/+]/*Rho*[+/-] retinas exhibited significantly better mean maximum amplitudes of scotopic a-waves (~37%) and photopic b-wave (~25%) at P30, as well as scotopic a-waves (~60%) and b-waves (~40%) at P90 (Fig. 1C, D).

For P17 *Prph2*[K153Δ/+]/*Rho*[+/-] mice, the assessment revealed modest non-significant functional improvements in the scotopic a- and b- as well as photopic b-wave mean maximum amplitudes compared to *Prph2*[K153Δ/+] (~14%, ~10%, and ~45%, respectively) (Fig. 1E, F). As the mice matured and their retinas fully developed, the partial ablation of *Rho* significantly enhanced scotopic a- and photopic b-wave amplitudes at P30 (~58% and ~51%, respectively). However, this statistically significant restorative effect was only apparent in the scotopic a-wave mean maximum amplitude (~42%) at P90 (Fig. 1E, F).

### Reduced RHO levels result in histologic improvements

To determine if the observed functional improvements were attributed to increased photoreceptor survival, we conducted outer nuclear layer (ONL) counts for the different genotypes at P30 and P90. Despite the functional improvements, histologic and morphometric analyses at the light level revealed only minor changes between the *Prph2* mutant models and their compound heterozygote/hemizygote counterparts (Fig. 2). At P30, ONL counts were similar among WT, *Rho*[+/-], *Prph2*[Y141C/+], and *Prph2*[Y141C/+]/*Rho*[+/-] (Fig. 2B, left). As the mice aged, significant loss of ONL nuclei was observed in *Rho*[+/-], *Prph2*[Y141C/+], and *Prph2*[Y141C/+]/*Rho*[+/-] compared to WT at P90 (Fig. 2B, right and see Table S4). Modest ONL loss was noted in *Prph2*[K153Δ/+] and *Prph2*[K153Δ/+]/*Rho*[+/-] at P30, progressing to significant photoreceptor loss in both models by P90 (Fig. 2C and see Table S5). However, a reduction in RHO levels was observed to improve average nuclear counts in the central regions of the *Prph2*[K153Δ/+]/*Rho*[+/-] retina at both time points (Fig. 2C).

### Reducing RHO levels leads to enhanced structural integrity of the outer segments

To explore additional factors contributing to the observed functional improvements, retinas underwent ultrastructural analyses. Low magnification ultrastructure images revealed that the elimination of one allele of *Rho* resulted in an overall enhancement of OS structure in both *Prph2* models (Fig. 3A, top panels). OSs display improved disc stacking and a reduction in the number of membranous whorls, particularly in *Prph2*[Y141C/+]/*Rho*[+/-] (arrowheads in Fig. 3A top panel).

Mutant *PRPH2*-associated defects in protein oligomerization have been linked to impaired disc closure, resulting in enlarged OS diameters and increased numbers of open, nascent discs[37]. Tannic acid staining allows differentiation between nascent and mature discs, as the exposed nascent discs exhibit a darker staining pattern due to their

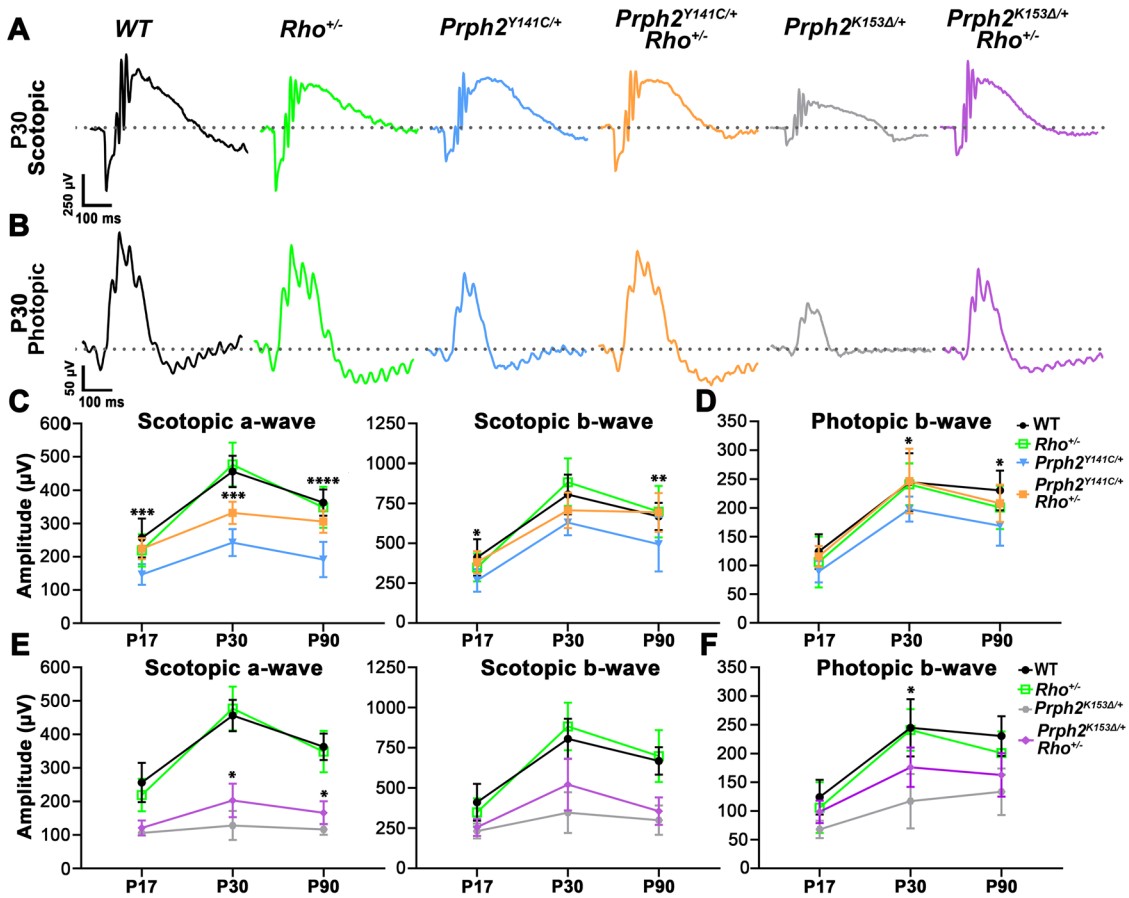

**Fig. 1 | Improved rod and cone functions in *Prph2^{Y141C/+}* and *Prph2^{K153Δ/+}* mice following partial ablation of *Rho*.** Representative scotopic (**A**) and photopic (**B**) ERG waveforms at P30 from all genotypes. **C**–**F** Mean maximum amplitudes of scotopic a-waves and b-waves, as well as photopic b-waves, plotted as mean ± SD at P17, P30, and P90. *P < 0.05, **P < 0.01, ***P < 0.001, ****P < 0.0001 by one-way ANOVA with Tukey's post-hoc comparison (see Table S3 for corresponding p values). N = 9–15 animals/genotype/age. *denotes comparisons between (**C**, **D**) *Prph2^{Y141C/+}* and *Prph2^{Y141C/+}/Rho^{+/-}* or (**E**, **F**) *Prph2^{K153Δ/+}* and *Prph2^{K153Δ/+}/Rho^{+/-}*.

exposure to the extracellular space[40]. Higher magnification images were utilized to measure OS diameters and count open discs, providing a quantitative assessment of the impact of RHO reduction on these morphological abnormalities (Fig. 3A, lower panels). In the presence of WT PRPH2 levels, removing one allele of *Rho* decreased the average number of open nascent discs from ~9 in WT to ~5 in *Rho^{+/-}* (Fig. 3B). Similarly, the reduction in RHO levels resulted in decreased numbers of open discs, with averages of ~18 for *Prph2^{Y141C/+}* and ~12 for *Prph2^{K153Δ/+}* reduced to ~6 and ~5 in *Prph2^{Y141C/+}/Rho^{+/-}* and *Prph2 ^{K153Δ/+}/Rho^{+/-}*, respectively (Fig. 3B). The rescue effect was also evident in OS diameter measurements (μm), with mean values of 1.2, 1.5, and 1.4 in WT, *Prph2^{Y141C/+}*, and *Prph2^{K153Δ/+}*, respectively, reduced to 1.0, 0.9, and 1.1 in *Rho^{+/-}*, *Prph2^{Y141C/+}/Rho^{+/-}*, and *Prph2^{K153Δ/+}/Rho^{+/-}* (Fig. 3C). These findings demonstrate that improved ultrastructure of OSs underlies the observed functional improvements in *Prph2^{Y141C/+}/Rho^{+/-}* and *Prph2^{K153Δ/+}/Rho^{+/-}* mice.

### Partial ablation of one allele of *Rho* leads to reduced levels of RHO relative to PRPH2

To establish the correlation between ablation of one *Rho* allele and the observed functional and structural improvements, immunodot blots were performed on retinal extracts from P30 mice. This aimed to determine the degree of decrease in RHO levels and the resulting RHO/PRPH2 ratio across all models (Fig. S1A). The partial genetic ablation of *Rho* resulted in reduction in protein levels in *Rho^{+/-}* (~56%), *Prph2^{Y141C/+}/Rho^{+/-}* (~55%), and *Prph2^{K153Δ/+}/Rho^{+/-}* (~65%) compared to WT, *Prph2^{Y141C/+}*, and *Prph2^{K153Δ/+}*, respectively (Fig. S1B). Notably, this

reduction did not impact the levels of PRPH2 (Fig. S1C). These findings highlight that the partial genetic ablation of *Rho* effectively decreases the ratio of RHO to PRPH2 in *Rho^{+/-}*, *Prph2^{Y141C/+}/Rho^{+/-}*, and *Prph2^{K153Δ/+}/Rho^{+/-}* retinas.

### Partial ablation of RHO does not impact the formation of higher-order complexes of PRPH2

Since proper PRPH2 oligomerization is essential for disc formation and OS structure, we examined the oligomerization status of PRPH2 in these mice following partial ablation of RHO[37,41]. Retinal extracts were subjected to immunoblotting after separation by sodium dodecyl sulfate-polyacrylamide gel electrophoresis (SDS-PAGE) under non-reducing conditions (Fig. S2A). Under these conditions, WT PRPH2 typically separates into monomers (~37 kDa) and disulfide-linked dimers (~75 kDa), while *Prph2^{Y141C/+}* and *Prph2^{K153Δ/+}* mutants have been shown to exhibit additional abnormal high-molecular weight protein complexes[13,14]. Our experimental results confirmed the presence of this abnormal complexes in *Prph2^{Y141C/+}/Rho^{+/-}* and *Prph2 ^{K153Δ/+}/Rho^{+/-}* retinas (Fig. S2A, red arrows). Quantification of the different complex forms was performed by measuring the signal intensity of each complex for PRPH2 and ROM1 and dividing it by the total signal intensity for the respective lane (Fig. S2B and C). No changes in the distribution of PRPH2 complexes were observed in either *Prph2^{Y141C/+}/Rho^{+/-}* (Fig. S2B, left) or *Prph2^{K153Δ/+}/Rho^{+/-}* (Fig. S2B, right) compared to their respective heterozygous mutant counterparts. These findings indicate that the reduction of RHO does not affect the oligomerization of PRPH2.

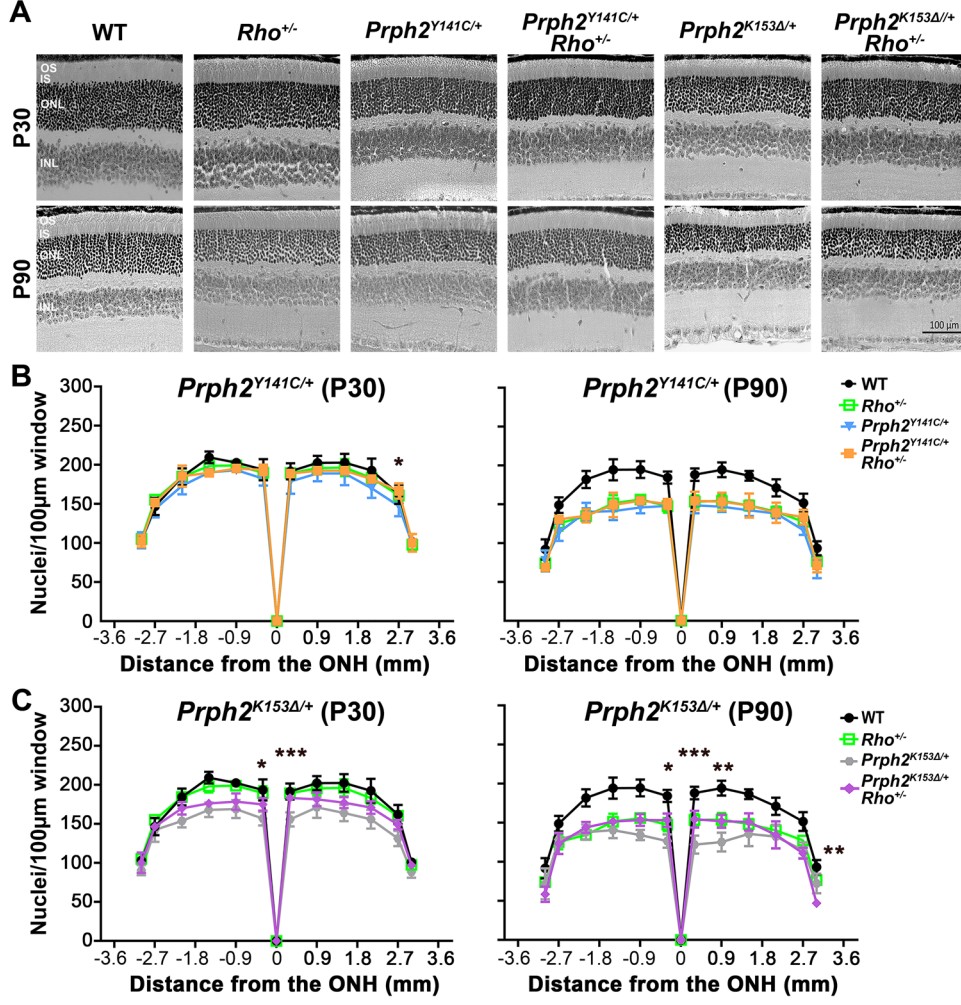

**Fig. 2 | Partial ablation of RHO does not affect ONL thickness in *Prph2*^K153Δ/+^ or *Prph2*^Y141C/+^. A** Representative images of H&E stained retinal sections at P30 and P90. **B**, **C** Nuclei counts were performed in 100 μm width windows in retinal sections taken from the indicated genotypes. Images were captured every 500 μm starting at the optic nerve head sections, and nuclei counts from multiple regions were plotted as mean ± SD for the listed genotypes (n = 3 for each genotype). *P < 0.05, **P < 0.01, ***P < 0.001 by two-way ANOVA (P30 *Prph2*^Y141C/+^: Interaction

P = 0.8626, Row Factor P < 0.0001, and Column Factor P < 0.0001. P90 *Prph2*^Y141C/+^: Interaction P = 0.0001, Row Factor P < 0.0001, and Column Factor P < 0.0001. P30 *Prph2*^K153Δ/+^: Interaction P = 0.0026, Row Factor P < 0.0001, and Column Factor P < 0.0001. P90 *Prph2*^K153Δ/+^: Interaction P < 0.0001, Row Factor P < 0.0001, and Column Factor P < 0.0001.) with Tukey's post-hoc comparison. *denotes comparisons between (**B**) *Prph2*^Y141C/+^ or (**C**) *Prph2*^K153Δ/+^and (**B**) *Prph2*^Y141C/+^/*Rho*^+/-^ or (**C**) *Prph2*^K153Δ/+^/*Rho*^+/-^. Scale bar corresponds to 100 μm.

## Reducing RHO level attenuates retinal gliosis in a mutation-independent manner

Müller cell gliosis, implicated in numerous retinal disorders[42], is characterized by the upregulation of glial fibrillary acidic protein (GFAP) as a non-specific marker of active gliosis and stress during retinal degeneration[43–45]. Since reducing RHO levels in both *Prph2* models led to functional and structural improvements, we next addressed the impact of that reduction on GFAP upregulation and its cellular distribution. Immunofluorescence and immunoblotting were conducted on P30 *Prph2*^Y141C/+^ and *Prph2*^K153Δ/+^ retinas in the presence of WT levels of RHO or after ablation of one allele (Fig. S3). In WT and *Rho*^+/-^ retinal sections, GFAP was predominantly localized in the nerve fiber layer consistent with previous literature (Fig. S3A, upper panels), whereas in *Prph2*^Y141C/+^ and *Prph2*^K153Δ/+^, GFAP extended to other retinal layers, including the ONL (Fig. S3A, arrows in middle left and lower left panels). Notably, the ablation of one *Rho* allele led to a reduction in retinal stress, evidenced by GFAP restriction to the nerve fiber, ganglion, and inner plexiform layers (Fig. S3A, arrows in middle right and lower right panels). However, quantification by immunodot blotting of P30 retinal extracts from the listed genotypes (Fig. S3B) revealed a non-statistically significant trend in GFAP reduction upon elimination

of one *Rho* complement (Fig. S3C). These findings collectively suggest that reducing RHO expression diminishes retinal stress associated with mutant *Prph2* expression, although further research is warranted to confirm this observation and its significance.

## Intravitreal treatment with *Rho*-specific ASO slows functional decline in *Prph2*^Y141C/+^ mice

We next investigated the potential of modulating RHO levels in a clinically relevant manner by the introduction of a rhodopsin specific ASO called *mRho* ASO1[38]. This ASO had already been tested and demonstrated efficacy in reducing endogenous mouse RHO[38]. Murray et al. showed "a dose-dependent reduction in rhodopsin mRNA was observed in eyes treated with *mRho* ASO1"[38]. Titrations of a single intravitreal injection of *mRho* ASO1 were performed at ages P15 and P45 to explore the dose-dependent effects of early and late-stage therapeutic intervention in *Prph2*^Y141C/+^ mice. Therapeutic efficacy was assessed by recording ERGs at ages P60 and P90 to identify changes in visual function. Results were expressed as percentages normalized to the vehicle control values for each dosage (Figs. S4 and S5). Due to procedure related complications with the smaller eyes of P15 mice, we report the initial titration results as the maximum amplitude (μV)

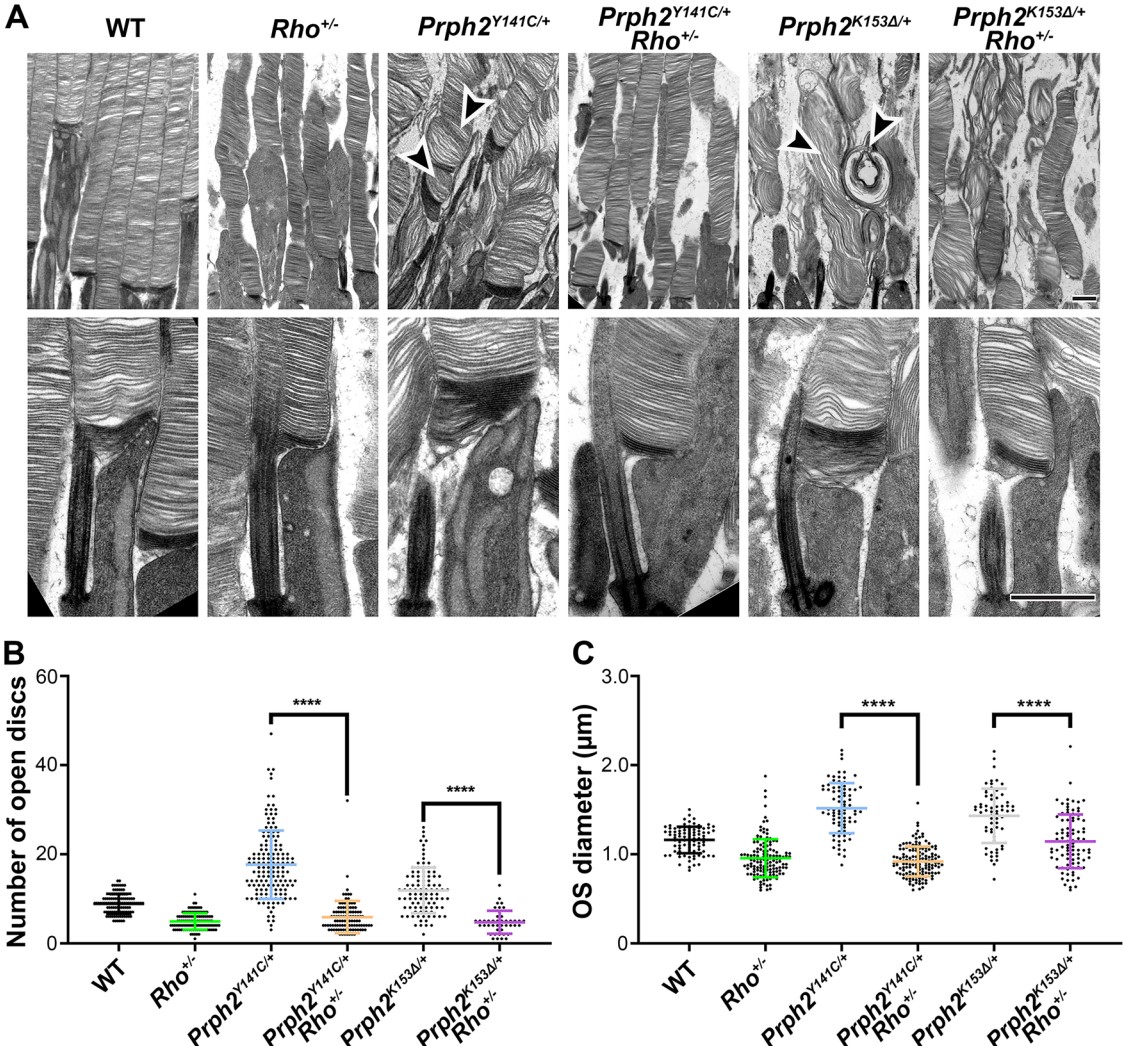

**Fig. 3 | *Prph2^{Y141C/+}/Rho^{+/-}* and *Prph2^{K153Δ/+}/Rho^{+/-}* exhibit improved OS morphogenesis and ultrastructure compared to their respective controls.**
**A** Representative TEM images at low and high-magnification of tannic acid/uranyl acetate-stained retinas from WT, *Rho^{+/-}*, *Prph2^{Y141C/+}*, *Prph2^{Y141C/+}/Rho^{+/-}*, *Prph2^{K153Δ/+}*, and *Prph2^{K153Δ/+}/Rho^{+/-}* at P16. **B** Quantification of open discs at the base of the OS in the listed genotypes at P16, utilizing data from a previous publication for WT and *Prph2^{Y141C/+}* open discs[37]. **C** Quantification of OS diameters measured in the listed genotypes at P16, with each data point representing a single outer segment.

*P < 0.05, **P < 0.01, ***P < 0.001, ****P < 0.0001 by one-way ANOVA (P < 0.0001 for both open disc and OS diameter) with Tukey's post-hoc comparison. Arrowheads highlight misaligned, overgrown, and sagittally aligned discs, as well as whorl-like structures commonly observed in *Prph2^{Y141C/+}* and *Prph2^{K153Δ/+}* ROSs. N (number of independent eyes used for analyses) values for (**B**): WT: 111, *Rho^{+/-}*: 112; *Prph2^{Y141C/+}*: 152; *Prph2^{Y141C/+}/Rho^{+/-}*: 105; *Prph2^{K153Δ/+}*: 98, *Prph2^{K153Δ/+}/Rho^{+/-}*: 43 and for C: WT: 86, *Rho^{+/-}*: 138, *Prph2^{Y141C/+}*: 80, *Prph2^{Y141C/+}/Rho^{+/-}*: 139, *Prph2^{K153Δ/+}*: 63, *Prph2^{K153Δ/+}/Rho^{+/-}*: 90. Scale bar, 1 μm. Error bars represent mean ± SD.

measured for independent *mRho* ASO1 treated samples divided by the mean maximum amplitude of independent vehicle injected controls (Fig. S4B–E). The results for P45 injected mice are reported as the maximum amplitude measured for the ASO treated eye divided by that of the contralateral vehicle injected control eye (Fig. S5B–E).

The titration experiments revealed that optimal dosages were age at intervention dependent. Following P15 intervention, a dose of 3.125 μg showed the most improvement at P60 as determined by maximum scotopic a- (~146%), scotopic b- (~137%), and photopic b- (~124%) amplitudes compared to vehicle control eyes (Fig. S4B and C). As the mice aged and the disease progressed, similar improvements were observed at P90 for both 1.56 μg (scotopic a-wave: ~167%, scotopic b-wave: 153%, and photopic b-wave: 120%) and 3.125 μg (scotopic a-wave: ~167%, scotopic b-wave: ~157%, and photopic b-wave: ~121%) treated eyes compared to their vehicle control eyes (Fig. S4D and E). Based on the observed increases in maximum amplitudes at P60 and P90 with the administration of 3.125 μg of *mRho* ASO1, we chose this dosage for the P15 intervention in the remainder of the study. As to the

intervention at P45, improvements in maximum amplitudes were observed at P60 following the administration of 6.25 μg (scotopic a-wave: ~135%, scotopic b-wave: ~117%, and photopic b-wave: ~105%) and 12.5 μg (scotopic a-wave: ~117%, scotopic b-wave: ~97%, and photopic b-wave: ~117%) dosages (Fig. S5B and C). The effects of the 6.25 μg dosage persisted at P90 (scotopic a-wave: ~133%, scotopic b-wave: ~112%, and photopic b-wave: ~108%), while the beneficial effects of the 12.5 μg dosage appeared to diminish (scotopic a-wave: ~92%, scotopic b-wave: ~80%, and photopic b-wave: ~105%) (Fig. S5D and E).

To demonstrate the specificity of this ASO, a control ASO (5'-CCTATAGGACTATCCAGGAA-3') was generated and used. We assessed the impact of intravitreal injecting the control ASO to ensure there were no adverse effects on retinal function. For this study, the most effective dose of 6.25 μg in 1 μl was used to evaluate the efficacy of the control ASO in P30 *Prph2^{Y141C/+}* mice. At 15 days post injection (PI), mice were evaluated functionally, structurally and biochemically (Figs. S6, S7 and S8, respectively). Results were compared to P45 old untreated and PI-15 vehicle injected animals. Functional assessments revealed

scotopic a- and b-waves, as well as photopic a- and b-waves, were indistinguishable from the untreated or vehicle treated controls (Fig. S6). Quantification of the ERG responses similarly showed no changes compared to the controls, both untreated and vehicle-injected (Fig. S6C). These results were further confirmed through light microscopy, and the number of photoreceptor nuclear counts showed no significant changes compared the control eyes (Fig. S7B and C). To verify that the injection of the control ASO did not affect the levels of endogenous RHO or PRPH2, immunodot blot analyses were conducted. Once again, no changes in the levels of RHO or PRPH2 were observed following the injection of the control ASO compared to untreated and treated samples (Fig. S8).

To further investigate our findings related to *mRho* ASO1 intervention at P15, we recorded ERGs from *Prph2*$^{Y141C/+}$ mice at PI-15, PI-45 and P75, respectively (Fig. 4A). Comparing ERG values for ASO-treated eyes with their untreated contralateral eyes revealed significant increases in mean treated maximal scotopic a-wave amplitudes at each evaluated time point (PI-15: ~107%, PI-45: ~112%, and PI-75: ~113%), while b-wave maximum amplitudes remained unchanged following therapeutic intervention (Fig. 4B, C). This pattern of specific improvement was also observed in photopic responses, where photopic a-wave maximum amplitudes were significantly improved at each time point of assessment compared to their corresponding control eye (PI-15: ~120%, PI-45: ~124%, and PI-75: ~130%), and no significant differences were observed when evaluating photopic b-wave maximum amplitudes (Fig. 4D).

Functional changes were evaluated at PI-15 and PI-45 following P45 *mRho* ASO1 intervention (Fig. 5A). Comparing ASO-treated eyes with their vehicle control contralateral counterparts at PI-15 revealed higher scotopic (~120%) and photopic (~144%) a-wave maximum amplitudes, while no observable differences were noted in scotopic and photopic b-wave maximum amplitudes (Fig. 5B, D). This improvement in a-wave amplitudes persisted at PI-45 (scotopic a-wave: ~124% and photopic a-wave: ~118%), with no enhancement observed in b-wave maximum amplitudes (Fig. 5C and D).

## ASO treatment lowers the ratio of RHO to PRPH2

Next, we explored the effects of ASO intervention on RHO and PRPH2 levels by quantifying protein amounts through immunodot blots and employing a two-tailed t-test for statistical analysis (Fig. 6A–F). The evaluation time points following intravitreal injection at P15 included ages P30, P60, and P90 corresponding to PI-15, PI-45, and PI-75. Treated eyes exhibited a reduction in mean RHO levels by ~32% at P30, and ~49% at P60, and P90, albeit this reduction was statistically insignificant at P30 (Fig. 6A). ASO administration at P15 did not significantly affect PRPH2 protein levels at any of the assessed time points (Fig. 6B). A statistically significant reduction in mean ratio of RHO to PRPH2 was observed at all three time points (P30: ~43%, P60: ~49%, and P90: 52%) (Fig. 6C). These improved ratios indicate the successful achievement of the desired therapeutic effect by efficiently reducing RHO levels. It is important to avoid excessive reduction of RHO, as it may exacerbate the degeneration and subsequently reduce PRPH2 levels.

Protein quantification was performed at PI-15 and PI-45 for intervention at P45 (P60 and P90, respectively). Average RHO protein levels showed a reduction of ~53% at P60 and ~71% at P90 compared to contralateral control eyes (Fig. 6D). Similar to the early intervention group, ASO treatment did not induce significant changes in average PRPH2 levels at both P60 and P90, although a slight decrease in the mean was observed at P90 (Fig. 6E). However, assessing the ration of RHO to PRPH2 revealed that the therapeutic effect of ASO treatment was successfully achieved, resulting in a decreased mean ratio of RHO to PRPH2 (P60: ~60% and P90: ~64%). Furthermore, this reduction was statistically significant at both time points (Fig. 6F).

To verify the mechanism of action for *mRho* ASO1 intervention, qRT-PCR was employed (primers used are in Table S1) to quantify *Rho*

and *Prph2* transcript levels post injection (Fig. 6G–I). Quantification of *Rho* transcript levels normalized to that of *Gapdh* revealed significant decreases in mRNA levels following injections at P15 (~35%) and P45 (~24%) (Fig. 6G). In contrast, evaluation of *Prph2* transcript levels normalized to *Gapdh* revealed no significant changes between uninjected and treated eyes following P15 injections. However, we observed a significant increase (~11%) in *Prph2* transcript levels following P45 injection of *mRho* ASO1 compared to vehicle control (Fig. 6H). We also assessed the relative abundance of the *Rho* transcript compared to *Prph2* transcript to gain insight into how the intervention affects this ratio. This revealed similar levels of significant reduction in the ratio of *Rho* to *Prph2* mRNA following P15 (~24%) and P45 (~31%) injections (Fig. 6I). These findings further support the mechanism of action of *mRho* ASO1 intervention in effectively reducing *Rho* transcript levels while exhibiting no significant impact on *Prph2* transcript levels.

## Age-dependent impact of *mRho* ASO1 intervention in mitigating histopathologic defects

To assess the histologic changes in *Prph2*$^{Y141C/+}$ following ASO treatment, morphometric analyses were conducted on retinal sections from P90 mice injected with either 3.125 μg *mRho* ASO1 at P15 or 6.25 μg *mRho* ASO1 at P45. Animals injected at P15 exhibited significant improvement in the number of photoreceptor nuclei compared to vehicle and untreated controls (Fig. 7A, B). However, injections at P45 did not show a significant improvement in number of photoreceptor nuclei, except for one area in the far superior periphery (Fig. 7C). When directly comparing the two intervention time points, it is evident that P15 intervention leads to improved ONL nuclear count throughout most of the retina, with a statistically significant improvement in the superior portion of the retina (Fig. 7D). Taken together, these results demonstrate the efficacy of early intervention using *mRho* ASO1 as a therapeutic strategy to delay photoreceptor death caused by the *Prph2*$^{Y141C/+}$ mutation.

## *mRho* ASO1 intervention improves OS ultrastructure

Ultrastructural examination of ROSs was performed 45 days post injection at both P15 and P45. ASO treated photoreceptors showed notable improvements in their upright appearance, organization and a reduction in whorl formations (Figs. S9, S10 and S11), which are characteristic features of mutant PRPH2 murine retinas[7,12,13,16,27,37]. Evaluating the percentage these whorls represent of the total ROSs throughout the retina revealed ~50% reduction after early ASO injections and ~62% reduction associated with late therapeutic intervention compared to contralateral controls (Fig. S11B, right panel). Additionally, numerous nucleated cells were frequently observed in the sub-retinal space of control P90 *Prph2*$^{Y141C/+}$ eyes, containing a large amount of phagocytosed material (Fig. S11A, arrow in right panel). While these cells were also present in ASO-injected animals, their presence was significantly reduced (compare Fig. S10A (arrowheads) to B). Evaluation of PBS-treated control *Prph2*$^{Y141C/+}$ eyes at P45 and collected at P90 revealed the presence of likely activated microglia. These cells were characterized by their nucleated appearance (Fig. S12A, arrowhead), localization next to the RPE (Fig. S12), extended processes (Fig. S12B, arrowhead), and the presence of phagocytized material and numerous lysosomes (Fig. S12C, arrowhead).

To further investigate the observed improvements in ROS ultrastructural, measurements of OS diameters and open disc counts were performed at PI-45 following both early and late treatment to quantitatively assess the known morphological defects (Fig. 8A). Treatment with *mRho* ASO1 resulted in a reduction in the number of open nascent discs compared to contralateral controls following both time points of intervention (P15 intervention: control ~12.4 and treated 7.1) (P45 intervention: control ~11.8 and treated ~7.6) (Fig. 8B and Table 1). This decrease was also evident in OS diameter measurements (P15 intervention: control ~1.8 μm and treated ~1.3 μm) (P45 intervention: control

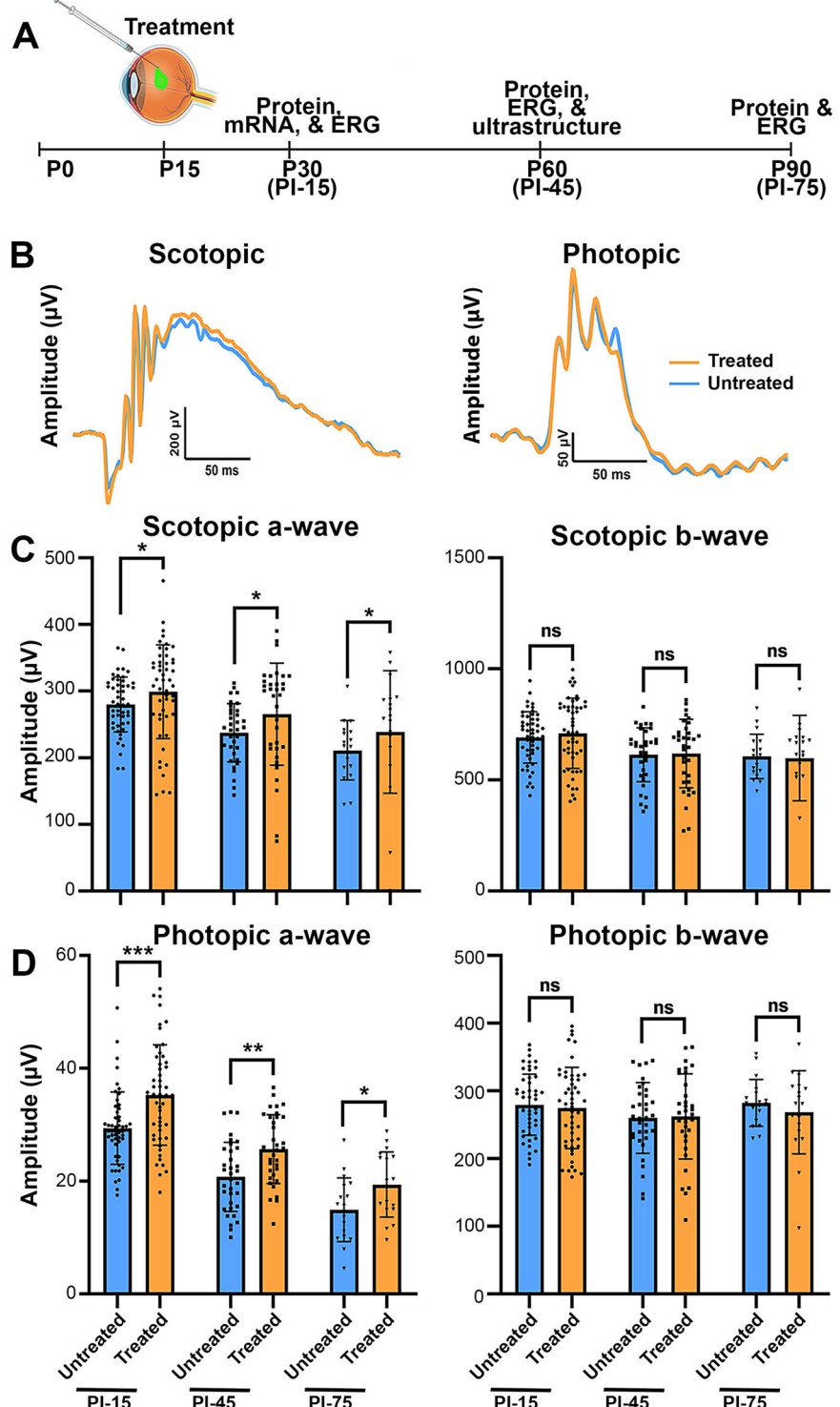

**Fig. 4 | Early-stage intervention with *mRho* ASO1 mitigates functional decline in *Prph2^{Y141C/+}* mice. A** Schematic representation of the design for early-stage preclinical intervention studies. Created with BioRender.com released under a Creative Commons Attribution-Non-commercial-NoDervis 4.0 International license. **B** Representative waveforms of scotopic and photopic responses recorded 15 days after injection at P30. Mean ± SD maximum amplitudes of scotopic a-waves and scotopic b-wave (**C**) and photopic a-wave and b-wave amplitudes (**D**) of treated (3.125 μg *mRho* ASO1) and untreated contralateral control eyes recorded in preclinical early injection studies at ages P30, P60, and P90. $^*P < 0.05$, $^{**}P < 0.01$, $^{***}P < 0.001$ by Mann-Whitney U test (PI-15: $N = 51$, PI-45: $N = 34$, and PI-75: $N = 18$).

~2 μm and treated ~1.3 μm) (Fig. 8C and Table 1). Similar to what was observed with genetic ablation, these results indicate that mitigation of these morphological defects plays a crucial role in driving the observed improvements in ROS ultrastructural following *mRho* ASO1 administration.

## Discussion

Photoreceptors exhibit high metabolic demands and are prone to the accumulation of toxic photo-oxidative products associated with the visual cycle. To ensure their proper function and overall health, photoreceptors undergo a daily physiological renewal process that is

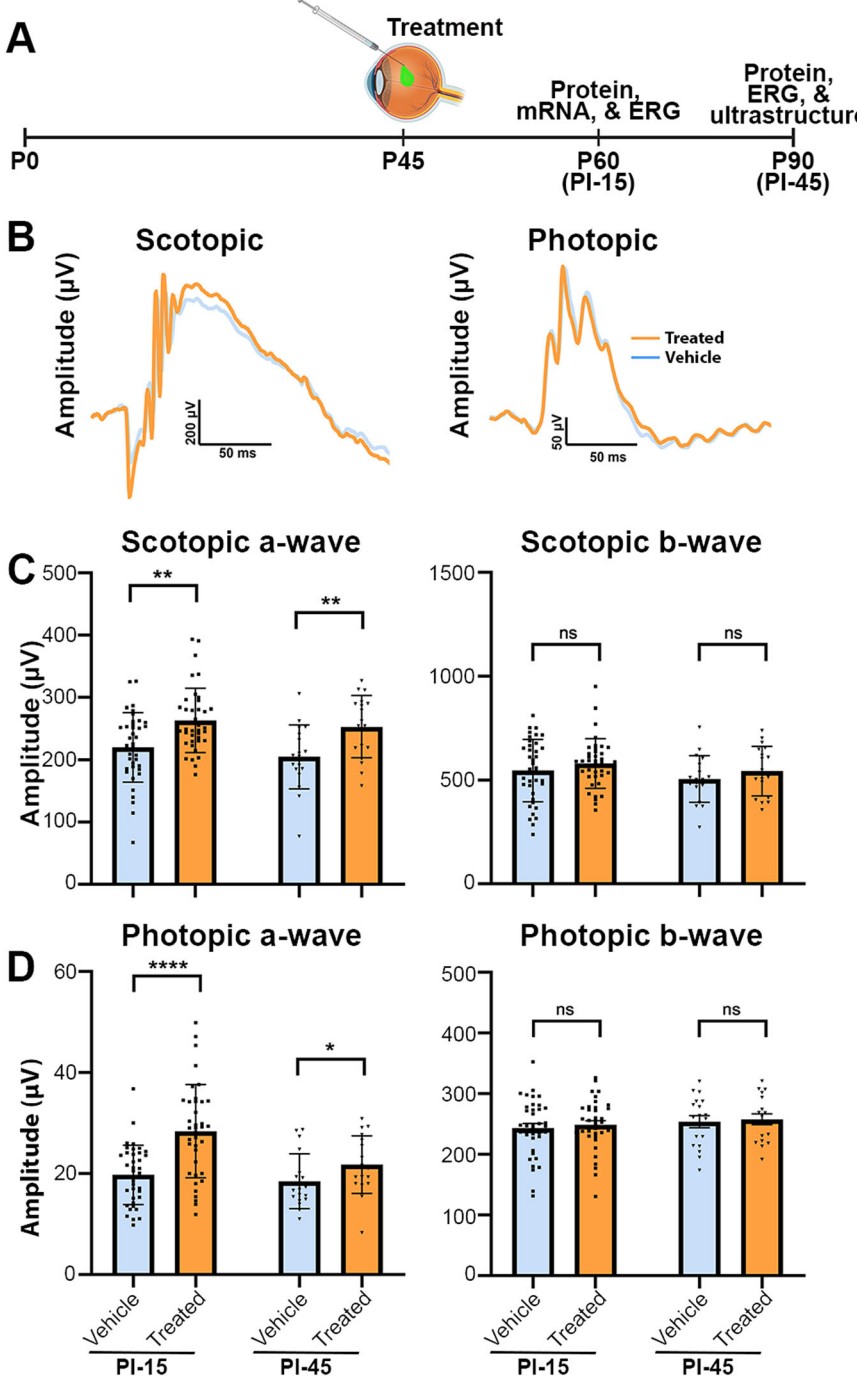

**Fig. 5 | Late-stage intervention with *mRho* ASO1 preserves functional performance in *Prph2^Y141C/+* mice. A** Design of late-stage preclinical intervention studies. Created with BioRender.com released under a Creative Commons Attribution-Non-commercial-NoDervis 4.0 International license. **B** Representative scotopic and photopic waveforms recorded at 15- and 45-days post injections at P45. **C, D** Mean maximum amplitudes of scotopic a-waves and b-waves, as well as photopic a-waves and b-waves were plotted as mean ± SD for treated eyes (6.25 µg *mRho* ASO1) and vehicle control contralateral eyes in the late-stage intervention studies conducted at P60 and P90. $^{*}P < 0.05$, $^{**}P < 0.01$, $^{***}P < 0.001$, $^{****}P < 0.0001$ by Mann-Whitney U test (PI-45: $N = 38$ and PI-75: $N = 18$).

crucial[46,47]. This process involves diurnal shedding of the distal portion of the outer segment, where discs are phagocytosed by the RPE, followed by the formation of new disc at the proximal end to replace them[46,47]. The presence of PRPH2 is crucial for proper disc morphogenesis, as its absence leads to failure in outer segment formation and the release of membrane evaginations from the connecting cilium in the form of ectosomes[27,48]. PRPH2 plays a dual role, not only inhibiting ectosome release but also participating in the proper enclosure of mature photoreceptor discs[37]. Defects in discs enclosure lead to misaligned overgrown discs and aberrations in the form of whorls, which are characteristics features observed in photoreceptor outer segments of animal models expressing *Prph2* pathogenic variants[13,14,16–18,49,50]. These structural abnormalities in the outer segments significantly impact photoreceptor function, leading to progressive loss of visual function and a reduction in the number of viable photoreceptor cells[13,14]. Despite the identification of numerous pathogenic variants in the *PRPH2* gene in patients[5,6], finding an effective therapeutic approach remains challenging.

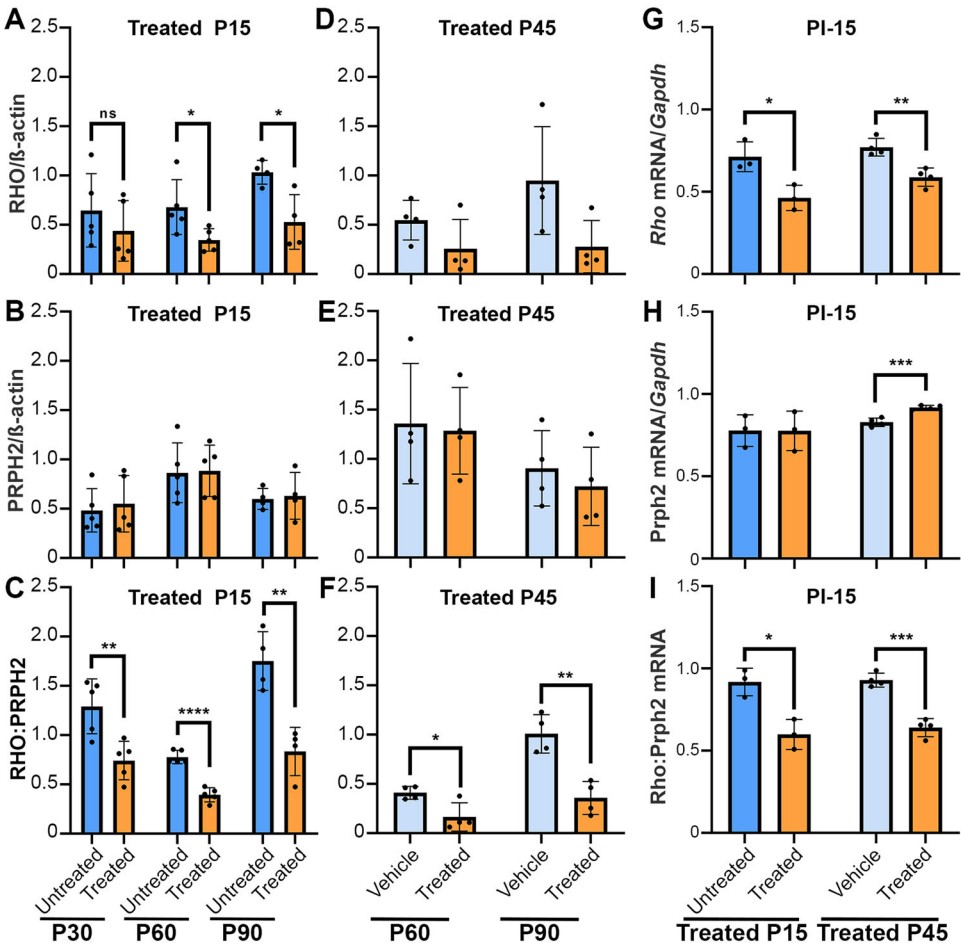

**Fig. 6 | Treatment with *mRho* ASO1 effectively reduced protein and transcript levels in *Prph2^{Y141C/+}* mice.** Quantification of (**A**) RHO and (**B**) PRPH2 in retinal extracts at P30, P60, and P90 after injections with *mRho* ASO1 at P15 (3.125 µg *mRho* ASO1). **C** Graphs depicting the ratio of RHO to PRPH2 following P15 injection, determined by dividing the RHO signal intensity value by that of PRPH2. Quantification of (**D**) RHO and (**E**) PRPH2 in retinal extracts at P60 and P90 after injections with *mRho* ASO1 at P45 (6.25 µg *mRho* ASO1). **F** Graphs illustrating the ratio of RHO to PRPH2 following P45 injections. **G–I** Transcript levels assessed by qRT-PCR 15 days post injection at either P15 or P45. Quantification of (**G**) *Rho* and (**H**) *Prph2* mRNA transcript levels relative to *Gapdh* for P15 and P45 injected samples. **I** Quantification of *Rho* mRNA transcript levels relative to *Prph2* for P15 and P45 injected samples. Data are presented as mean ± SD. $^*P < 0.05$, $^{**}P < 0.01$, $^{***}P < 0.001$, $^{****}P < 0.0001$ as determined by student t-test analysis. $N = 3$-5 retinas per treatment condition.

In pursuit of finding approaches to delay disease progression in a comprehensive manner, we explored a novel therapeutic strategy targeting the ratio between RHO and PRPH2 to address pathological structural phenotype. Mutations in *Prph2* often lead to reduced levels and/or functionality of PRPH2, disrupting the essential photoreceptor protein balance needed for cellular homeostasis. Given that the abundance of RHO influences the size and structure of rod outer segment discs[26,28], we hypothesized that reducing RHO levels in *Prph2* mutant models could mitigate the phenotypic overgrowth abnormalities.

In proof of concept studies, we utilized partial genetic ablation of *Rho* to modulate protein levels in two well-established mouse models expressing mutant PRPH2, *Prph2^{Y141C/+}* and *Prph2^{K153Δ/+}*[13,14]. This reduction in RHO levels resulted in improved maximum physiological responses driven by rods in both models due to enhanced rod outer segment structure. Interestingly, despite being a rod-specific protein, the decrease in RHO reduction also improved cone-driven responses in both models. This finding is in line with the well-documented symbiotic relationship between rods and cones, where cone degeneration is often secondary to rod degeneration in retinitis pigmentosa[51,52]. The proper alignment of rod outer segment may contribute to the upright positioning of cone outer segments, resulting in enhanced responses to light stimulation. Previous studies have demonstrated that rods

produce factors supportive of cones[53,54]. It is conceivable that *mRho* ASO1 treatment improved rod outer segment structures, thereby reducing cellular stress and promoting the production of these supportive factors, consequently leading to improved cone responses.

Although we observed only transient improvements in physiological function for *Prph2^{K153Δ/+}*, the potential of RHO reduction as a therapeutic strategy for this mutation is promising. Previous efforts to supplement the *Prph2* gene in *Prph2^{K153Δ/+}* mice by crossing them with a PRPH2-overexpresser line demonstrated limited efficacy, resulting in minimal improvements in scotopic a-wave ERGs at P30[13]. Therefore, the transient rescue of scotopic a-, scotopic b-, and photopic b-wave maximum amplitudes associated with RHO reduction is particularly exciting. One possible explanation for this transient rescue is based on our observations from ASO titration studies, suggesting that levels of RHO reduction may offer therapeutic benefits in the early stages but become detrimental as the disease progresses. In future studies, it will be crucial to explore whether controlled modulation of RHO levels in *Prph2^{K153Δ/+}* mice can result in sustained therapeutic benefits.

To demonstrate the clinical relevance of this strategy and achieve controlled reduction of RHO levels, we opted to use a previously characterized ASO, *mRho* ASO1[38]. This ASO was employed as a control by Murray et al. and demonstrated specificity for wild type RHO[38]. ASOs are single-stranded nucleic acids with chemically modified

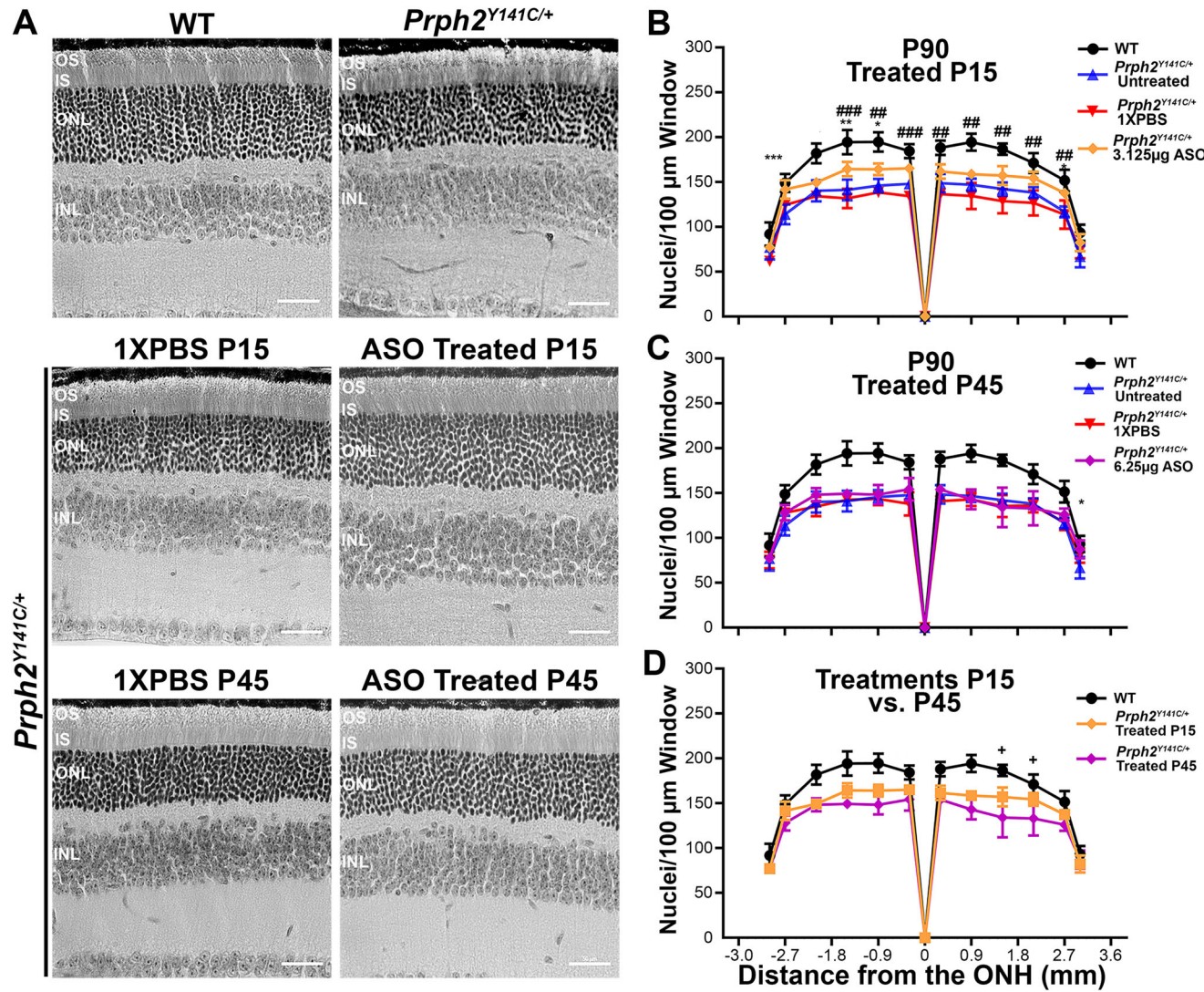

**Fig. 7 | Early-stage treatment with *mRho* ASO1 successfully delays ONL thinning in *Prph2*^*Y141C/+* mice. A** Representative images of retinal sections stained with H&E at P90. **B**–**D** Nuclear counts from 100 μm-windows at every 500 μm distance from the optic nerve and across the superior-inferior plane of retinal sections collected from P90 un-injected, 1X PBS injected as a control and *mRho* ASO1 injected animals following (**B**) early (P15) or (**C**) late-stage (P45) therapeutic intervention. **B**, **C** WT and *Prph2*^*Y141C/+* controls were added for comparison. **B**, **C** *$P < 0.05$, **$P < 0.01$, ***$P < 0.001$ by two-way ANOVA (P15 Injected: Interaction $P < 0.0001$, Row Factor $P < 0.0001$, and Column Factor $P < 0.0001$. P45 Injected: Interaction $P < 0.0001$,

Row Factor $P < 0.0001$, and Column Factor $P < 0.0001$. P15 vs. P45 Injected: Interaction $P < 0.0001$, Row Factor $P < 0.0001$, and Column Factor $P < 0.0001$.) with Tukey's post-hoc comparison. *denotes comparisons between *Prph2*^*Y141C/+* and (**B**) *Prph2*^*Y141C/+* 3.125 μg ASO or (**C**) *Prph2*^*Y141C/+* 6.25 μg ASO. #denotes comparisons between (**B**) *Prph2*^*Y141C/+* 3.125 μg ASO and P15 injected *Prph2*^*Y141C/+* 1X PBS. **D** Plotted are mean ± SD values from (**B**) and (**C**) for WT, *Prph2*^*Y141C/+* P15 Injected 3.125 μg ASO, and P45 injected 6.25 μg ASO *Prph2*^*Y141C/+* for ease of comparison. + denotes comparisons P15 injected and P45 injected. *N* = 3 animals for all genotypes and experimental conditions. Scale bar corresponds to 50 μm.

backbones designed to bind to their target mRNA and regulate protein expression[55]. Traditionally, this heteroduplex formation aims to decrease aberrant protein expression through various mechanisms, including RNase H mediated cleavage of the heteroduplex, splicing modulation, and steric hindrance of ribosomal binding[55,56]. ASOs have shown a good safety profile in numerous clinical trials, resulting in 10 FDA approved therapeutics[56]. The specific oligonucleotide, *mRho* ASO1, has the added benefit of previously demonstrating in vivo efficacy in reducing *Rho* mRNA in a dose-dependent manner following intravitreal injection[38]. Intravitreal administrations have become a standard procedure for the treatment of retinal diseases since the first FDA approved intravitreal injection therapeutic in 1998[57]. With millions of intravitreal treatments performed annually, significant advancements have been made in injection-related procedures and tools to minimize patient discomfort and injection complications, making

intravitreal injection the current preferred method of delivery for retinal disease therapy[58].

To explore the therapeutic window of our approach, we evaluated two intervention time points, considering the wide variability in age of onset and severity for *PRPH2* associated phenotypes[6]. For these experiments, we selected *Prph2*^*Y141C/+* mice, known for their slower rate of degeneration, allowing for a longer window of intervention and assessment[7,14,59]. Early intervention was initiated shortly after mice open their eyes at P15, while late intervention took place at P45 after full development of the retina and the initiation of functional decline in rods and cones[7,14,59]. Both early and late therapeutic interventions resulted in improved scotopic and photopic maximum a-wave responses. While it was expected that earlier intervention would yield superior functional rescue, the maximum a-wave values were similar between early and late intervention when evaluated at P60 and

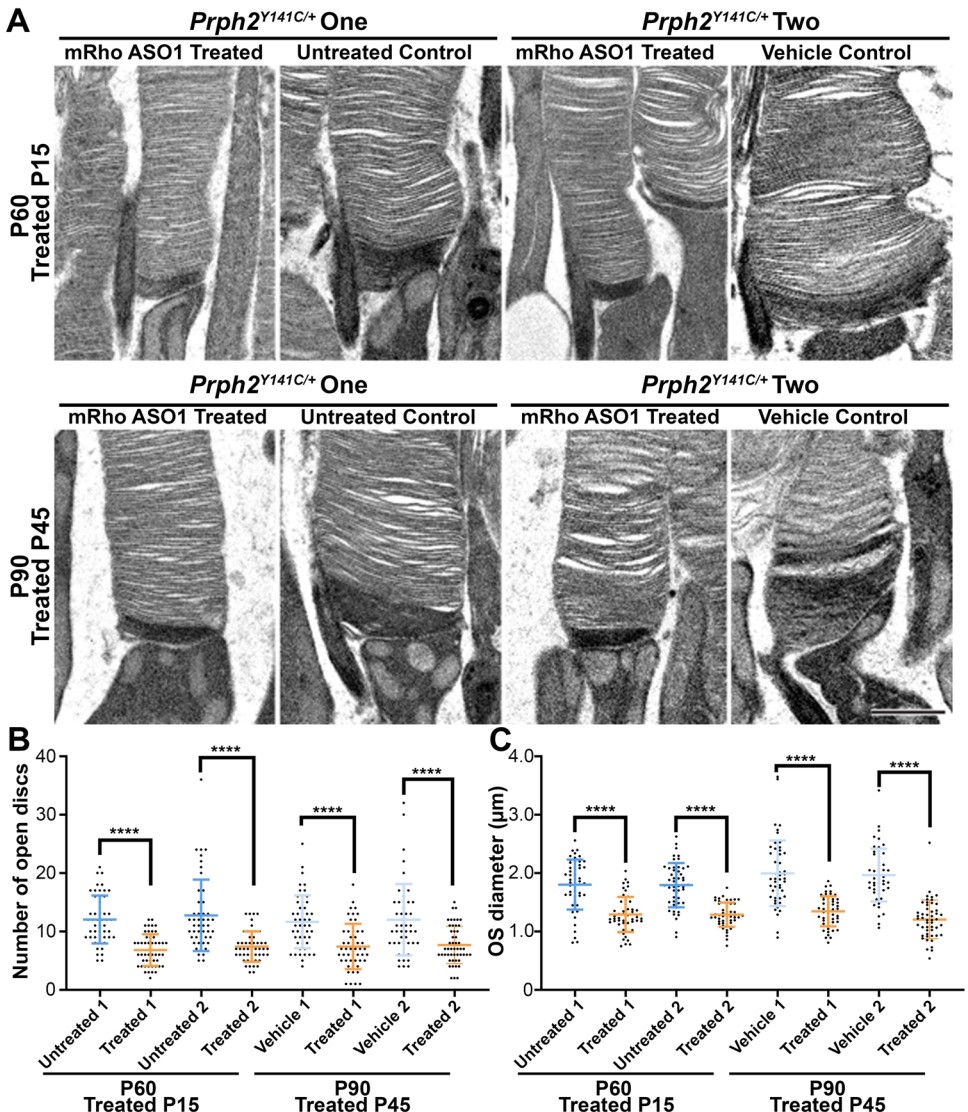

**Fig. 8 | Injection of *mRho* ASO1 led to significant improvements in the morphogenesis and ultrastructure of OSs in *Prph2*^Y141C/+ mice. A** Representative high-magnification TEM images of tannic acid/uranyl acetate-stained retinas taken from P90 *mRho* ASO1 treated and contralateral control eyes that were injected at P15 (3.125 μg *mRho* ASO1) and P45 (6.25 μg *mRho* ASO1). Quantification of open discs at the base of the OS (**B**) and OS diameters (**C**) in *mRho* ASO1 injected and contralateral control eyes 45 days after treatment. Each data point represents a single OS. *$P < 0.05$, **$P < 0.01$, ***$P < 0.001$, ****$P < 0.0001$ by one-way ANOVA ($P < 0.0001$ for both open discs and OS diameter) with Tukey's post-hoc comparison. $N = 50$ OSs per each sample for (**B**) and (**C**). Scale bar, 1 μm. Error bars represent mean ± SD.

P90. This could possibly be due to a waning effect of the therapeutic benefit over time, as P15 intervention was administered 30 days earlier. Future studies involving repeated dosage administration may provide more insight into this aspect. However, no significant changes in maximum b-wave amplitudes were observed following *mRho* ASO1 treatment, despite observing improvements in b-wave responses following partial genetic ablation of RHO in *Prph2*^Y141C/+ mice.

In addition to the observed improvements in ERG responses, early *mRho* ASO1 intervention led to significant increases in the number of photoreceptor nuclei. By administering the *mRho* ASO1 to modulate protein levels, we were able to optimize RHO reduction to achieve the proper RHO:PRPH2 ratios, resulting in maximum physiological response improvements. This dosage preserved the photoreceptors and slowed down the rate of degeneration. An inadequate reduction of RHO would lead to insignificant changes and minimal preservation, while excessive reduction of RHO would worsen photoreceptor death and degeneration. Late stage *mRho* ASO1 intervention failed to delay photoreceptor cell death. It is possible that, despite optimization for maximum functional improvement, P45 might be beyond the

threshold of disease progression for *mRho* ASO1 to effectively improve photoreceptor survival. These findings further underscore the importance of maintaining proper RHO:PRPH2 ratios and choosing the appropriate time point for intervention to effectively prevent photoreceptor death using the reduction of RHO as a strategy.

In line with our observations in the *Prph2* double mutants, ultrastructural analyses revealed that *mRho* ASO1 therapy improved rod outer segment ultrastructure throughout the retina, as evidenced by a reduction in large membranous whorls and sagittally oriented discs. Additionally, reduction of RHO following *mRho* ASO1 treatment reduced the structural abnormalities of increased open nascent discs and outer segment diameters associated with *Prph2* mutations. Furthermore, we observed a potential decrease in microglia infiltration following late stage *mRho* ASO1 intervention at P90. As microglia infiltration is often observed in photoreceptor degenerative conditions[60–62], this decrease further illustrates the overall benefit of *mRho* ASO1 on overall photoreceptor health and structure.

Taken together, these results signify the therapeutic efficacy of reducing levels of RHO as a strategy in impeding retinal degeneration

**Table 1 | Quantification of open discs and OS diameter for P15 and PI-45 *Prph2*^Y141C/+ *mRho* ASO treated and their corresponding untreated contralateral control eyes**

| Samples | Number of Open Discs (mean ± SD) | OS Diameter (μm) (mean ± SD) |
|---|---|---|
| **P15 *Prph2*^Y141C/+ injected mice with *mRho* ASO1** | | |
| P15 Contralateral Control One | 12.0 ± 0.6 | 1.80 ± 0.06 |
| P15 Treated Sample One | 6.8 ± 0.4 | 1.29 ± 0.04 |
| P15 Contralateral Control Two | 12.8 ± 0.9 | 1.79 ± 0.05 |
| P15 Treated Sample Two | 7.4 ± 0.4 | 1.29 ± 0.03 |
| **P45 *Prph2*^Y141C/+ injected mice with *mRho* ASO1** | | |
| P45 Contralateral Control One | 11.7 ± 0.6 | 2.00 ± 0.08 |
| P45 Treated Sample One | 7.4 ± 0.5 | 1.35 ± 0.04 |
| P45 Contralateral Control Two | 12.0 ± 0.9 | 1.97 ± 0.06 |
| P45 Treated Sample Two | 7.7 ± 0.5 | 1.21 ± 0.05 |

caused by *PRPH2* pathogenic variants. However, while these findings are exciting, there are limitations to consider when translating this approach into clinical relevance. The *Prph2*^Y141C/+ knockin model has been shown to recapitulate many known patient phenotypes, including photoreceptor structural defects, decreased scotopic and photopic ERG values, and fundus flecking[14]. However, differences between human and mouse retinas hinder complete replication of the underlying disease mechanisms[63–65]. Additionally, genetic variations, variable penetrance, and diverse age-of-onset observed in humans[19–24,66] present challenges in determining which patient would benefit from therapeutic intervention. Nonetheless, the strategy of reducing RHO levels holds promise as a preventative therapeutic approach for individuals with *PRPH2* mutations, although long-term investigations in healthy control animals would be necessary before implementation.

Despite these limitations, these studies present significant findings that we aim to explore further in the future. We have introduced a distinctive strategy with the potential to yield therapeutic benefits across various *PRPH2* pathogenic variants by targeting critical protein ratios essential for outer segment morphogenesis and structural stability. Future investigations will focus on expanding the application of this strategy to include additional *Prph2* mutations, exploring the long-term effects of *Rho* knockdown in healthy controls, and assessing the feasibility of repeated dosing in long-term *Prph2* mutants.

## Methods

### Study approvals and generation/acquisition of mutant lines

All animal experiments were approved by the University of Houston Institutional Animal Care and Use Committee (IACUC) and adhered to NIH guidelines and the recommendations from the Association for Research in Vision and Ophthalmology (Rockville, MD).

The knockin mouse models, *Prph2*^K153Δ/K153Δ and *Prph2*^Y141C/Y141C, were generated and characterized previously[13,14]. Additionally, an in-house WT mouse strain was generated as described previously[67]. The homozygous *Rho* knockout mouse line (*Rho*^−/−) was originally obtained as a generous gift from Dr. Janis Lem (Tufts University, Boston, MA, USA) and has been previously described[34]. Knockin heterozygotes (*Prph2*^K153Δ/+ and *Prph2*^Y141C/+) mice were generated by crossing the WT mice with either *Prph2*^K153Δ/K153Δ or *Prph2*^Y141C/Y141C mice, respectively. Double heterozygotes (*Prph2*^K153Δ/+/*Rho*^+/− and *Prph2*^Y141C/+/*Rho*^+/−) were produced through crossbreeding of *Prph2*^K153Δ/K153Δ or *Prph2*^Y141C/Y141C mice with *Rho*^−/− mice. All mice were raised under cyclic light conditions, with a 12h-light/12-dark cycle (12 L/12D), and an illuminance of 30 lux. Mice were fed *ad libitum* mouse diet 9 F (5020) from LabDiet

(Quakertown, PA, USA)). Mice were euthanized by carbon dioxide asphyxiation followed by cervical dislocation. Both mouse sexes were used in this study and no differences were observed between them.

The p.Tyr141Cys variant, identified as (c.422 A > G), has been previously characterized in patients[22,23]. Our Y141C knockin mouse model precisely carries the same mutation, Y141C (c.422 A > G), and has been well characterized[7,14,68]. The p.Lys154del variant, identified as (c.461-463del), was initially reported as 3-bp deletion at position 153/154[19]. The deletion affects one of the two conserved lysines in both human and mouse PRPH2 at positions 153 and 154, corresponding to c.458-460 and c.461-463 in human (see below). The human cDNA sequence at this site is AAG-AAG, and patients have a deletion of one of the AAG. The mouse cDNA at this site is AAA-AAG. To mirror the human condition, the K153Δ knockin model was generated by deleting AAA at position 153 site, resulting in a mutant protein lacking on of the two lysines. As a result, this mouse line aligns with both the initial patient report referring to the deletion of 153/154[19], and the updated information in the Leiden Open Variation Database (LOVD) and in recent literature, where the mutation is identified as c.461_463del, p.Lys154del (db-ID PRPH2_000084)[69]. This mouse line, K153del, has been previously characterized[13].

### Human PRPH2

cDNA: 428-478 458 461
 GAC ACA GAC ACC CCT GGC AGG TGT TTC ATG **AAG AAG** ACC ATC GAC ATG CTG
 D T D T P G R C F M **K K** T I D M L
AA: 143-159

### Mouse Prph2

cDNA:428-478 458 461
 GAT ACG GAC ACC CCC GGC CGG TGC TTC ATG **AAA AAG** ACC ATC GAC ATG CTC
 D T D T P G R C F M **K K** T I D M L
AA: 143-159

### Electroretinography

Full field scotopic and photopic electroretinograms (ERGs) were recorded as described before[70], with minor modifications using a UTAS system (LKC; Gaithersburg, MD). Eyes of overnight dark-adapted mice were dilated with 1% cyclopentolate solution (Bausch+Lomb; Bridgewater, NJ) prior to inducing anesthesia via intramuscular injection of 85 mg/kg ketamine (Covetrus; Portland, ME) and 14 mg/kg xylazine (Akorn, Inc.; Lake Forest, IL). The reference needle electrode was placed subdermally between the ears instead of the cheek as previously described[70]. Values obtained for ERG experiments using the double heterozygote animals were averaged and plotted. For injected animals, following the recording of the initial values, the probes were replaced, and a second measurement was taken to ensure observed trends were accurate. Only the initially recorded values were used for analysis.

### Histology and morphometry

Histology and morphometric analyses were performed as previously described[67]. Briefly, eyes were enucleated from euthanized mice and fixed in modified Davidson's fixative[71] (12% formaldehyde, 15% ethanol, and 5% glacial acetic acid) overnight at 4 °C. Fixed eyes were then dehydrated and embedded in paraffin before being sectioned at 10 μm thickness. Sections containing the optic nerve head were stained with hematoxylin and eosin (H&E) (Sigma Aldrich, Burlington, MA, MHS16 and HT110116) and mounted with Permount (Fisher Scientific, Waltham, MA, P15-100) mounting medium. For morphometric analysis, 100 μm width window images were captured every 500 μm starting at the optic nerve head using a Zeiss Axioskop and a 20X objective lens. Image J was used for morphometric analysis by manually counting the

number of nuclei in the 100 μm window images captured at the indicated distances. Representative images were taken using a 40X objective.

## Protein extraction and quantification

Retinas from the indicated genotypes and treatment groups were extracted from euthanized mice and immediately flash frozen using liquid nitrogen before being stored at -80 °C until processing. Retinas were homogenized and then sonicated in 50 μL 1X RIPA Buffer (ABCAM, Boston, MA, ab156034) per retina. Samples were then incubated in the same extraction buffer over night at 4 °C. Following extraction, samples were centrifuged at 18,200 × g for 15 minutes at 4 °C to remove insoluble debris. Extracted protein concentrations were measured using the colorimetric Bradford Assay (Bio-Rad, Hercules, CA, #5000006).

Immunodot blots were used to quantify protein levels. Extract titrations were performed to determine the optimal range of total protein level needed from each genotype prior to quantification experiments. A MINIFOLD I microsample filtration manifold (Whatman® Schleicher & Schuell®, Keene, NH #27510) was used to load the protein extracts onto a nitrocellulose membrane (Bio-Rad, Hercules, CA #1620112). The membrane was allowed to dry before blocking with 5% non-fat milk in 1X TBST (0.1% Tween®). Following blocking, membranes were incubated with either unconjugated primary antibody (anti-RHO, -PRPH2, or -GFAP) or anti-actin primary antibody conjugated with Horseradish Peroxidase (HRP; Table S2) at room temperature. Membranes were washed prior to incubation with the appropriate HRP-conjugated secondary antibodies at room temperature. Following a final wash, the membrane was incubated with ECL reagent (SuperSignal™ West PICO PLUC Chemiluminescent Substrate #34577) for 1 minute, chemiluminescence imaging was performed using ChemiDoc™ MP imaging system (Bio-Rad, Hercules, CA), and quantification was performed using Image Lab software v6.0.1 (Bio-Rad).

For protein quantification using immunodot blot, three to five replicates were used for each genotype and for each treatment. Each sample was independently measured in triplicate on each blot, averaged to be considered one value and presented as a mean + SD. For all immunoblotting quantification involving double heterozygote mice, chemiluminescent signal intensity values obtained for PRPH2, RHO, and GFAP were normalized by β-actin signal intensity before further normalization to the internal WT control. Quantification for the ASO treatment experiments involved normalization of measured RHO and PRPH2 chemiluminescent signal intensity values by β-actin signal intensity to obtain RHO/β-actin and PRPH2/β-actin values. Additionally, RHO was normalized by PRPH2 in order to obtain RHO:PRPH2 ratios.

Protein oligomerization was analyzed by performing non-reducing sodium dodecyl sulfate-polyacrylamide gel electrophoresis (SDS-PAGE) followed by immunoblotting using previously described methods[14]. Chemiluminescent signal intensity values obtained for each band were divided by the total lane signal intensity to obtain the percent distribution.

## Transmission electron microscopy (TEM)

Fixation and processing of mouse eyes for TEM was performed as described previously[40]. In short, anesthetized mice were transcardially perfused with 2% paraformaldehyde, 2% glutaraldehyde and 0.05% calcium chloride in 50 mM MOPS (pH 7.4). Eyes were post-fixed for an additional 2 hour. Eyecups were embedded in 2.5% low-melt agarose (Precisionary Instruments, Greenville, NC) and sectioned on a Vibratome (VT1200S; Leica, Buffalo Grove, IL). 200 μm agarose sections were stained with 1% tannic acid (Electron Microscopy Sciences, Hatfield, PA) and 1% uranyl acetate (Electron Microscopy Sciences). Stained sections were gradually dehydrated with ethanol and

infiltrated and embedded in Spurr's resin (Electron Microscopy Sciences). 70 nm plastic sections were cut, placed on copper grids and stained with 2% uranyl acetate and 3.5% lead citrate (19314; Ted Pella, Redding, CA). Samples were imaged on a JEM-1400 electron microscope (JEOL, Peabody, MA) at 60 kV with a digital camera (BioSprint; AMT, Woburn, MA). Image analysis and processing was performed with ImageJ.

## *mRho* ASO1 synthesis

A previously characterized *Rho*-specific ASO *mRho* ASO1 targeting exon 5 (5′-AGCTACTATGTGTTCCATTC−3′) are chimeric 20-mers with phosphorothioate backbone containing 20-O-methoxyethyl (MOE) modification at positions 1-5 and 15-20 was used in this study[38]. The control (5′ CCTATAGGACTATCCAGGAA- 3′) and *mRho* ASO1, were prepared by Integrated DNA Technologies, Inc. (Coralville, Iowa) using identical chemistry[38]. Synthesis was performed according to company's standard operating procedures using proprietary processes. The control ASO has been previously employed as a control in various contexts: to suppress polypyrimidine tract binding protein 1 in neurons[72], to increase microglial phagocytosis[73], to deliver to the cornea[74], to alleviate hepatic fibrosis[75], in finding a protective factor in Dravet Syndrome[76], in knockdown of GADD34 to ameliorate ALS[77], to inhibit mTORC2[78] and to activate human frataxin expression[79].

## Intravitreal Injections

Intravitreal injections were performed as previously described[80]. Briefly, *Prph2*[Y141C/+] mice were injected intravitreally at P30 for control ASO, P15 for early-stage intervention and P45 for late-stage intervention. Eyes were dilated with 1% cyclopentolate solution (Bausch + Lomb; Bridgewater, NJ) for 5 minutes prior to intramuscular injection of 85 mg/kg ketamine (Covetrus; Portland, ME) and 14 mg/kg xylazine (Akorn, Inc.; Lake Forest, IL). Adequate anesthesia was confirmed prior to injection by pedal withdrawal reflex assessment. A 30 gauge sterile needle was used to puncture the cornea and provide a guide hole. A Hamilton syringe was inserted through the guide hole to manually dispense material [*mRho* ASO1 or Endotoxin-Free Dulbecco's PBS (1X) (EMD Millipore; Billerica, MA)] into the vitreal chamber. Volumes injected for all studies were 0.5 μL in P15 animals and 1.0 μL in P45 animals. Dosages used for P15 titration studies included 1.5625 μg, 3.125 μg, 6.25 μg, 12.5 μg, and 25 μg, while dosages used for P45 titrations included 3.125 μg, 6.25 μg, 12.5 μg, 25 μg, and 50 μg. Following dosage titration studies, a concentration of 6.25 μg/μL was used for all injections so that P15 mice received 3.125 μg *mRho* ASO per injection while P45 mice received 6.25 μg *mRho* ASO per injection. Following injection, Triple antibiotic ointment (Taro pharmaceuticals Inc., Hawthorne, NY, USA) was applied to each eye. All animals were closely monitored during recovery from anesthesia until fully ambulatory and provided adequate access to food and water. Mice experiencing injection related complications, such as significant accumulation of blood in the vitreous chamber, retinal detachment, damage to the iris or lens, intraocular infection, or cataract development were excluded from the study. All injections were performed in a sterile surgical suite. In the titration studies, both eyes were injected for all mice with either *mRho* ASO1 or 1X PBS vehicle control. Due to procedure related difficulties experienced during the titration studies when attempting to inject both eyes at P15, all characterization studies were performed injecting one eye with 3.125 μg ASO and the other serving as an uninjected contralateral control. All P45 characterization studies were performed using the same procedure as titration studies with 6.25 μg ASO.

## Immunofluorescence

Immunofluorescence was performed as previously described[59]. Briefly, mice were euthanized by carbon dioxide asphyxiation followed by cervical dislocation, eyes were enucleated, and then fixed in modified

Davidson's fixative overnight at 4 °C[71]. Fixed eyes were then dehydrated and embedded in paraffin before being sectioned at 10 µm thickness. Before use, samples were incubated in xylene to remove paraffin and tissue was rehydrated in sequential ethanol dilutions. Following rehydration, antigen retrieval was performed by incubating samples in sodium citrate at 100 °C for 30 minutes before being placed in blocking buffer (5% bovine serum albumin, 1% donkey serum, and 0.5% Triton X-100 in PBS) for 1 hour. Following blocking, antigen labeling was performed by incubating sections with primary antibodies overnight, washing with 1X PBS, incubating with secondary antibodies for 2 hours, and then washing with 1X PBS. DAPI staining was then performed for 15 minutes at a concentration of 0.1 µg/mL (Thermo Fisher Scientific, Waltham, MA). Slides were mounted using anti-fade mounting media and sealed with clear nail polish. Imaging was performed using a 20x objective on a Zeiss LSM800 confocal microscope (Zeiss, White Plains, NY). Image processing was performed in Zen 3 lite software.

### Antibodies
All antibodies used for immunoblotting and immunofluorescence are listed in Supplementary Table 2.

### Statistical analysis
Statistical Analysis was performed using GraphPad Prism version 8 (GraphPad Software, La Jolla, CA). One-way ANOVA with Tukey's post-hoc comparisons was employed for quantitative analysis between groups for all double mutant ERG data and comparison of protein levels in double mutant figures. Two-way ANOVA with Tukey's post-hoc comparisons was employed for quantitative analysis between groups for all morphometry analysis. ASO treated ERGs were compared using two-tailed Mann–Whitney U test. Student's two-tailed t-test was used for statistical analysis of protein quantification studies. Significance was set at $P < 0.05$. Pairwise comparison significances are indicated as $*P < 0.05$, $**P < 0.01$, $***P < 0.001$, $****P < 0.0001$ throughout the text.

### Reporting summary
Further information on research design is available in the Nature Portfolio Reporting Summary linked to this article.

## Data availability
All data associated with this study are presented in the paper or the Supplementary Materials. Source data are provided with this paper Source data are provided with this paper.

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

## Acknowledgements

All authors acknowledge Larissa Ikelle and Drs. Lars Tebbe and Shannon Conley for their thoughtful review and valuable insights on the manuscript. This work was supported by grants from the NIH/NEI [EY010609 (MIN and MRA), EY034671 (MIN), EY030451 (VYA), EY005722 (VYA), and EY033763 (TRL). NIH had no role in the collection, analyses, or interpretation of the data; in the writing of the manuscript or in the decision to submit the manuscript for publication.

## Author contributions

C.R.W., M.I.N and M.R.A designed the study. C.R.W. gathered and executed most of the experiments, analyzed the data and wrote the first draft of the manuscript. M.S.M performed and analyzed the P17 and P30 double heterozygote ERG studies, nuclear counts, intravitreal injections of the control ASO, and the blots in Fig. S2. R.C. performed intravitreal injections of the Rho-ASO and participated in the ASO titration experiments. SZ was responsible for obtaining and analyzing the control ASO data presented in Figs. S6, S7 and S8. T.R.L., C.M.C. and V.Y.A were responsible for all of the EM experiments and the data presented in Figs. 3, 8, S9, S10, S11, and S12 and Table 1, M.K. performed GFAP IHC presented in Fig. S3A, MIN and MRA edited the manuscript to its final form.

## Competing interests

The authors declare no competing interests.
