## [Peer Review File · Nature Communications]

Downregulation of rhodopsin is an effective therapeutic strategy in ameliorating peripherin-2-associated inherited retinal disordersREVIEWER COMMENTS

Reviewer #1 (Remarks to the Author):

This is an interesting paper dealing with mouse models of inherited retinal degeneration caused by mutations in the PRPH2 gene, for which there are no currently available therapies. In this study, the authors explore the possibility that reducing the expression of the rhodopsin gene may have therapeutic benefit in cases where the PRPH2 gene is mutated, based on the hypothesis that the ratio of prph2 to rhodopsin is critical for photoreceptor morphology. The advantage of this approach is that one treatment methodology knocking down rhodopsin could be applicable to many different PRPH2 mutations. A disadvantage is that this treatment would not resolve any issues resulting from direct effects of PRPH2 mutations on cones; however, it could potentially reduce “bystander” cone death. The treatment approach is addressed in two different modeling systems, first, by breeding animals with PRPH2 mutations with rhodopsin knockouts to generate animals with both a PRPH2 mutation and heterozygous loss of rhodopsin, and second by using intravitreal injection of an antisense oligonucleotide designed to reduce rhodopsin expression (modeling a possible therapeutic approach that could be applied to human patients). Two different PRPH2 mutations are examined in the context of rhodopsin knockout (Y141C and K153Δ), and only one mutation is examined in the context of antisense treatment (Y141C). The authors demonstrate that both approaches result in improved photoreceptor morphology relative to animals with PRPH2 mutations only, and improved electrophysiological function of the retina (by electroretinography). In the reviewer’s opinion, the strongest result is the recovery of rod OS morphology: the rhodopsin knockout substantially improves the morphology of the retinas with PRPH2 mutations. A caveat for these experiments is that while N is high in terms of photoreceptors examined, it is low (perhaps 2 per condition?) in terms of animals examined. However, a low N is largely unavoidable in experiments based on electron microscopy. There are further caveats: In experiments with rhodopsin knockouts, the effect on the ERG is substantial (Figure 1), while the effect on cell death is minimal (figure 2B, 90 day timepoint, results are not significant). In particular, the scotopic a wave shifts from ~230 to ~330 uV in figure 1c at the p30 timepoint with P given as “*****” (if I’m not mistaken, actually not defined in legend, but presumably even lower than <0.001 given for “***”). In contrast, with antisense treatment

(figure 4c) the size of the shift is much smaller, with $P < 0.05$, despite a very large number of data points ($N = 51$). However, for the antisense treatment cell death data, there is a substantial effect at the 90 day timepoint across the retina, particularly when comparing the animals treated with antisense therapy to animals treated with PBS carrier (the comparison to animals that received no treatment is less significant). Thus, it seems that knockout has a greater effect on the ERG and less effect on cell death, while the opposite is true for antisense therapy, although the two approaches were not directly compared in an experiment. One possibility is that the treatment with antisense PBS vehicle exacerbates the retinal degeneration, making the observed differences between treatment and sham treatment larger. Another issue is that, while the overall effects of the antisense treatment on rhodopsin expression were examined by western blot, there is no measure of transfection efficiency, or the area of the retina transfected (for example, by fluorescently tagging the antisense oligo or in-situ detection by other means), or cell-to-cell variability in knockdown. Finally, the functional recovery is measured only by ERG, and it is not clear that this represents improved vision. Ultimately, blindness in human patients is caused by long term progressive cell death, and this study does not examine long term efficacy. However, that said, overall, I found the paper very interesting, and the data is high quality, addressing a condition for which there is no current treatment. However, the size of the effect of treatment is small and without long term data it is not clear that this is a viable treatment strategy; therefore, there should be caution on the part of the authors in stating any conclusions in this regard. Below are more specific comments:

Abstract: the abstract begins referring to “Peripherin-2”, and then switches to “peripherin”. Is this referring to the gene vs gene product, if so it could be clarified. If not, naming should be kept consistent to avoid confusion. Moreover, Prph2 is used in the body of the paper.

Figure 1 and associated text: the statistical analysis for this figure is complex. The statistical comparisons shown are (I believe) post-hoc comparisons following a 2-way ANOVA, in which the post-hoc comparison is within a given age-group (e.g p30), and across genotypes, using the Tukey post-hoc test. However, in addition, the three p-values from the 2-way ANOVA test should be given, as the post-hoc test is only applicable in the case of a significant ANOVA result.

The associated text notes that “Peak responses were observed at P30, corresponding to full retinal maturation, followed by an aged-related decline at P90. Notably, a small increase in the mean photopic b responses was observed for Prph2K153Δ/+ between P30 (117μV) and P90 (134 μV) (Fig. 1, A-F).”

However, P values for the type of comparison described are not given. Was there an effect of age by two-way ANOVA, and were subsequent post-hoc tests of individual genotypes across ages (such as between Prph2K153Δ/+ at P30 vs P90 described in the text above) statistically significant, if yes, what are the p values.

In the paragraph at the top of page 6, a similar claim is made:

“However, by P90, this restorative effect diminished as the mice aged (42%, 19%, and 22%, respectively), with no statistically significant changes in scotopic or photopic maximum amplitude values.”

The word “changes” suggests change over time, in which case P-values are not given.

However, I believe in this case the authors are actually referring to comparisons within an age group, and could substitute text such as “...this restorative effect was not apparent in older P90 mice, with no statistically significant differences in scotopic or photopic maximum amplitude values.” (in particular, substituting the word “differences” for “changes”).

Figure 2 and associated text: The text referring to Figure 2 is a paragraph under the heading “Reducing RHO levels in knockin models of Prph2 mutations results in improvements in the structural integrity of the outer segments.” However, this paragraph of text, and the data in Figure 2, do not refer to the structural integrity of outer segments, but rather cell death measured via nuclear counts, with subsequent paragraphs under this heading expanding on structural integrity data from Figure 3. I recommend giving this paragraph its own heading. In contrast, the parallel experiment involving antisense treatment is given its own heading, and refers to this as “histopathologic defects”. This is a minor issue and only became apparent when I was scrolling back through the paper’s headings to find the section on cell death. Similar to figure 1, P-values for the 2-way ANOVA should be reported (and for all other 2-way ANOVAs).

Figure 3: very interesting data, no comments other than those above.

Figure 4: ASO treatment effects on the ERG: it is odd that the most significant result is the PI-15 photopic ERG (as this assay measures cone function, and cones do not express rhodopsin). This conundrum is mentioned in the discussion as being outside the scope of the study. One possibility that I believe is supported by the data: the ERG assay may be measuring the misalignment of photoreceptors. If the rod and cone OS are not as parallel (more disorganized) in the untreated PRPH2 mutants, this would reduce the ERG a wave signal, which is due to summation of aligned electrical signals. In contrast, the B-wave derived from the inner retina is unaffected as these cells are not misaligned; the data indicates they are receiving very similar inputs from treated and untreated photoreceptors.

The ERG analysis appears to be based on a very large number of data points (for example, for one condition, $N = 51$). If these represent individual animals this figure represents a heroic amount of intravitreal injection and data collection. Is N actually the number of individual animals?

Figure 5: Similar questions apply: why is there is a large effect on cones, and are the data points distinct animals?

Figure 6: it is interesting that there is a much higher level of significance to the rho:prph2 ratio relative to the rho:actin ratio. Possibly, this is because all rho and prph2 are derived from rod OS, while the actin levels are measured over many cell types?

Figure 7: again, please give the significance levels for the 2-way ANOVA.

Figure 8: very interesting data showing antisense oligonucleotide treatment improves OS morphology, supported by additional supplemental figures.

Supplemental data:

Figure S1: confirmatory evidence that partial rho knockout reduces rho:prph2 ratio. No issues.

Figure S2: a negative result, but an interesting experiment: does knockdown of rho alter the formation of large complexes or aggregates associated with the two prph2 mutants? It does not. No issues.

Regarding figure S3, which examines reactive gliosis: "These findings collectively suggest that reducing RHO expression diminishes retinal stress associated with mutant Prph2 expression." However, the statistical analysis of the dot blots returned a non-significant result. Perhaps this could be qualified with "although further confirmatory research is warranted" or similar.

S4-5: titration of oligonucleotide levels used to determine the doses used in Figs 4-8. No issues.

S6-S8: lower magnification images and image analysis showing effects of antisense treatment on OS morphology supporting the experiment shown in figure 8

Supplemental tables: table of QPCR primers, table of antibodies, table of P values associated with retinal morphometry. I did not review the table of morphometry values, authors please double-check as this is a very complicated table.

Reviewer #2 (Remarks to the Author):

The authors of the manuscript entitled "Downregulation of rhodopsin is an effective therapeutic strategy in ameliorating peripherin-2-associated inherited retinal disorders" described a potential novel and innovative therapeutic approach to slow down the progression of the disease caused by genetic defects in the *PRPH2* gene.

This is a very innovative approach that might become a potential treatment in the future for patients suffering from this disease. Especially because ASOs are easy to deliver and well tolerated in the eye. Thus, these findings are very promising to further develop this strategy.

Major comment:

The manuscript show very beautiful data and it is really well written and easy to follow. My main concern is the lack of proper controls for the ASO experiments. The authors have used an ASO previously characterized to downregulate RHO expression. In this context the authors are claiming that by downregulating RHO, the phenotype of mouse models carrying mutations in the *Prph2* gene improve the phenotype. However, it is not discriminated if the delivery of an ASO itself, or the chemical modifications of this ASO can be the inducers of this improvement. For that, members of the Oligonucleotide Therapeutics Society published in open access the guidelines on which controls are needed for the different types of ASOs (<https://www.liebertpub.com/doi/10.1089/nat.2018.0772>). This guidelines have been fully accepted in the field and are very helpful to design experiments. This guidelines were developed after several ASOs showing phenotype improvement that later it was shown to be recapitulated with a non-targeting control ASO. Indicating, that the effect was not directly related to the targeted gene. It is for that reason that the vehicle itself is not enough as a control. Therefore, authors need to perform the experiments with at least one control oligo with similar properties to show that the delivery of any ASO does not improve the phenotype.

Minor comments:

It is not accepted anymore that when referring to humans or disease the word “mutation” is used. Please make sure they are change to variants, pathogenic variants, genetic defects, etc. It is still accepted for mouse, plasmids, etc.

The first time the variants are given, please provide also the c.XXX nomenclature together with the protein.

In the introduction, the part in which the ASO is mentioned seems to come completely out of the blue. Perhaps some more context would improve readability.

Although the ASO is published, please indicate the sequence and chemical modification in the methods.

Please indicate the amount of ASO injected in all the figure legends that relate to assessing a parameter upon ASO delivery. This applies to main figures 4, 5, 6, 8; and supplementary figure S6.

Sometimes results are given in mean \pm SD and others mean \pm SEM, can the authors elaborate on the reason for that?

Response to reviewers' comments

We appreciate the time dedicated by the reviewers and value their comments and suggestions. All feedback has been thoroughly addressed in the revised version of the manuscript. Below, you will find our responses to each of the reviewer's comments:

Reviewer #1 (Remarks to Authors and responses):

A caveat for these experiments is that while N is high in terms of photoreceptors examined, it is low (perhaps 2 per condition?) in terms of animals examined. However, a low N is largely unavoidable in experiments based on electron microscopy.

Answer: We agree with the reviewer that a sample size of N=2 for electron microscopy is unavoidable due to cost and effort.

There are further caveats: In experiments with rhodopsin knockouts, the effect on the ERG is substantial (Figure 1), while the effect on cell death is minimal (figure 2B, 90 day timepoint, results are not significant). In particular, the scotopic a wave shifts from ~230 to ~330 uV in figure 1c at the p30 timepoint with P given as "*****" (if I'm not mistaken, actually not defined in legend, but presumably even lower than <0.001 given for "****").

Answer: We agree with the comment of this reviewer regarding the significant improvements on the ERG but not on the cell death. We have defined the P values in the revised figure legend.

In contrast, with antisense treatment (figure 4c) the size of the shift is much smaller, with P <0.05, despite a very large number of data points (N = 51). However, for the antisense treatment cell death data, there is a substantial effect at the 90 day timepoint across the retina, particularly when comparing the animals treated with antisense therapy to animals treated with PBS carrier (the comparison to animals that received no treatment is less significant). Thus, it seems that knockout has a greater effect on the ERG and less effect on cell death, while the opposite is true for antisense therapy, although the two approaches were not directly compared in an experiment. One possibility is that the treatment with antisense PBS vehicle exacerbates the retinal degeneration, making the observed differences between treatment and sham treatment larger.

Answer: While it is desirable to compare the results of genetic ablation to ASO injection, it is not feasible due to fundamental differences. Genetic ablation involves germline modification, leading to knockin retinal development under reduced RHO levels, whereas ASO injections occur postnatally, with the earliest injection at P15. These distinctions may account for the observed slight differences in our results. Regarding the issue of untreated versus vehicle-treated animals, we do observe a slight reduction in ERG with the vehicle treatments.

Another issue is that, while the overall effects of the antisense treatment on rhodopsin expression were examined by western blot, there is no measure of transfection efficiency, or the area of the retina transfected (for example, by fluorescently tagging the antisense oligo or in-situ detection by other means), or cell-to-cell variability in knockdown. Finally, the functional recovery is measured only by ERG, and it is not clear that this represents improved vision. Ultimately, blindness in human patients is caused by long term progressive cell death, and this study does not examine long term efficacy.

Answer: The ASO used in our experiments has been previously published in *Murray et al (IOVS 2015: 6362-6375)*, demonstrating its even distribution throughout the retina following intravitreal

injection. While the assessment of ASO distribution, along with the mentioned experiments, is planned for the near future, the current manuscript's focus is on proving the concept. The aim is to inspire further experimentation, hastening the translation of this approach to the clinic. We, along with the reviewers, recognize the urgency of finding cures for retinal degenerative diseases caused by pathogenic variants in the *Prph2* gene.

However, that said, overall, I found the paper very interesting, and the data is high quality, addressing a condition for which there is no current treatment.

Answer: We express our gratitude for the positive comments provided this reviewer.

However, the size of the effect of treatment is small and without long term data it is not clear that this is a viable treatment strategy; therefore, there should be caution on the part of the authors in stating any conclusions in this regard.

Answer: We do agree with this comment, as long-term analysis and multiple deliveries of ASOs are integral components of the planned experiments mentioned above.

Below are responses to the specific comments:

Abstract: the abstract begins referring to “Peripherin-2”, and then switches to “peripherin”. Is this referring to the gene vs gene product, if so it could be clarified. If not, naming should be kept consistent to avoid confusion. Moreover, *Prph2* is used in the body of the paper.

Answer: This has been corrected in the revised version.

Figure 1 and associated text: the statistical analysis for this figure is complex. The statistical comparisons shown are (I believe) post-hoc comparisons following a 2-way ANOVA, in which the post-hoc comparison is within a given age-group (e.g p30), and across genotypes, using the Tukey post-hoc test. However, in addition, the three p-values from the 2-way ANOVA test should be given, as the post-hoc test is only applicable in the case of a significant ANOVA result.

The associated text notes that “Peak responses were observed at P30, corresponding to full retinal maturation, followed by an aged-related decline at P90. Notably, a small increase in the mean photopic b responses was observed for *Prph2K153Δ/+* between P30 (117μV) and P90 (134 μV) (Fig. 1, A-F).”

However, P values for the type of comparison described are not given. Was there an effect of age by two-way ANOVA, and were subsequent post-hoc tests of individual genotypes across ages (such as between *Prph2K153Δ/+* at P30 vs P90 described in the text above) statistically significant, if yes, what are the p values.

In the paragraph at the top of page 6, a similar claim is made: “However, by P90, this restorative effect diminished as the mice aged (42%, 19%, and 22%, respectively), with no statistically significant changes in scotopic or photopic maximum amplitude values.”

The word “changes” suggests change over time, in which case P-values are not given. However, I believe in this case the authors are actually referring to comparisons within an age group, and could substitute text such as “...this restorative effect was not apparent in older P90 mice, with no statistically significant differences in scotopic or photopic maximum amplitude values.” (in particular, substituting the word “differences” for “changes”).

Answer: We appreciate the reviewer's comments and the opportunity to rectify an error related to the statistical analysis. Originally, we used a 2-way ANOVA, treating the presence/absence of a *Prph2* mutation and full complement/partial genetic ablation of *Rho* as independent variables.

Following consultation with a statistician regarding values from the 2-way ANOVA results, we learned that genotype should be considered as a single independent variable. Consequently, we revised the analysis using a 1-way ANOVA, and the corresponding figures and texts have been updated to reflect these changes.

Figure 2 and associated text: The text referring to Figure 2 is a paragraph under the heading “Reducing RHO levels in knockin models of Prph2 mutations results in improvements in the structural integrity of the outer segments.” However, this paragraph of text, and the data in Figure 2, do not refer to the structural integrity of outer segments, but rather cell death measured via nuclear counts, with subsequent paragraphs under this heading expanding on structural integrity data from Figure 3. I recommend giving this paragraph its own heading. In contrast, the parallel experiment involving antisense treatment is given its own heading, and refers to this as “histopathologic defects”. This is a minor issue and only became apparent when I was scrolling back through the paper’s headings to find the section on cell death. Similar to figure 1, P-values for the 2-way ANOVA should be reported (and for all other 2-way ANOVAs).

Answer: We have implemented the suggested changes in this section to enhance clarity. These paragraphs have been separated into two sections as the reviewer recommended. We have included the P-values for the 2-way ANOVA in the figure legend as requested.

Figure 3: very interesting data, no comments other than those above.

Answer: Thank you for your kind comment.

Figure 4: ASO treatment effects on the ERG: it is odd that the most significant result is the PI-15 photopic ERG (as this assay measures cone function, and cones do not express rhodopsin). This conundrum is mentioned in the discussion as being outside the scope of the study. One possibility that I believe is supported by the data: the ERG assay may be measuring the misalignment of photoreceptors. If the rod and cone OS are not as parallel (more disorganized) in the untreated PRPH2 mutants, this would reduce the ERG a wave signal, which is due to summation of aligned electrical signals. In contrast, the B-wave derived from the inner retina is unaffected as these cells are not misaligned; the data indicates they are receiving very similar inputs from treated and untreated photoreceptors

Answer: We have incorporated a section in the discussion that briefly discusses cone improvement, considering the possibility brought up by the reviewer. The following paragraph was added “The proper alignment of ROS may contribute to the upright positioning of cone OSs, resulting in enhanced responses to light stimulation. Previous studies have demonstrated that rods produce factors supportive of cones {Kelley, 2017 #77;Leveillard, 2004 #78}. It is conceivable that ASO treatment improved ROS, thereby reducing cellular stress and promoting the production of these supportive factors, consequently leading to improved cone responses.”

The ERG analysis appears to be based on a very large number of data points (for example, for one condition, N = 51). If these represent individual animals this figure represents a heroic amount of intravitreal injection and data collection. Is N actually the number of individual animals?

Answer: The sample size, denoted as N, represents the number of individual animals, with each data point corresponds to an individual eye. In the treatment group, one eye of each animal is plotted, while the contralateral eye is represented in the control group. We appreciate the reviewer’s recognition of our team’s diligent efforts in conducting intravitreal injections and collecting data for this manuscript.

Figure 5: Similar questions apply: why is there is a large effect on cones, and are the data points distinct animals?

Answer: Please see our response above to Figure 4 regarding the effect on cone function. N represents the number of individual animals, with each data point corresponding to an individual eye. In the treatment group, one eye of each animal is plotted and the contralateral eye is represented in the control group.

Figure 6: it is interesting that there is a much higher level of significance to the rho:prph2 ratio relative to the rho:actin ratio. Possibly, this is because all rho and prph2 are derived from rod OS, while the actin levels are measured over many cell types?

Answer: We found this observation intriguing during manuscript preparation. We agree that it might be a result of the OS specificity of RHO and PRPH2, which aligns with the improvements observed in OS ultrastructure following treatment.

Figure 7: again, please give the significance levels for the 2-way ANOVA.

Answer: We have included the P-values for the 2-way ANOVA in the figure legend as requested.

Figure 8: very interesting data showing antisense oligonucleotide treatment improves OS morphology, supported by additional supplemental figures.

Answer: Thank you for your kind comment. No modifications have been made to Figure 8.

Supplemental data:

Figure S1: confirmatory evidence that partial rho knockout reduces rho:prph2 ratio. No issues.

Answer: No changes have been made to Figure S1.

Figure S2: a negative result, but an interesting experiment: does knockdown of rho alter the formation of large complexes or aggregates associated with the two prph2 mutants? It does not. No issues.

Answer: The knockdown of RHO did not induce observable changes in the formation of large complexes or aggregates associated with the two *Prph2* mutants. Therefore, no modifications are made to this figure.

Regarding figure S3, which examines reactive gliosis: “These findings collectively suggest that reducing RHO expression diminishes retinal stress associated with mutant Prph2 expression.” However, the statistical analysis of the dot blots returned a non-significant result. Perhaps this could be qualified with “although further confirmatory research is warranted” or similar.

Answer: We have incorporated the requested qualifier into the text: “These findings collectively suggest that reducing RHO expression diminishes retinal stress associated with mutant Prph2 expression, although further confirmatory research is warranted to confirm this observation.”

S4-5: titration of oligonucleotide levels used to determine the doses used in Figs 4-8. No issues.

Answer: No changes made to Fig. S4-5.

S6-S8: lower magnification images and image analysis showing effects of antisense treatment on OS morphology supporting the experiment shown in figure 8

Answer: No changes made to Fig. S6-8.

Supplemental tables: table of QPCR primers, table of antibodies, table of P values associated with retinal morphometry. I did not review the table of morphometry values, authors please double-check as this is a very complicated table.

Answer: The morphometry tables were directly imported from the GraphPad Prism multiple comparisons results output, ensuring the accuracy of the data. We opted for this approach to avoid cluttering the spider plots with symbols of significance between various genotypes and distances from the optic nerve. Including the table allows readers the opportunity to gain a more in-depth understand of the statistical analyses, which could not be fully incorporated into the figure without making the figure overly crowded.

Reviewer #2 (Remarks to the Author):

The authors of the manuscript entitled "Downregulation of rhodopsin is an effective therapeutic strategy in ameliorating peripherin-2-associated inherited retinal disorders" described a potential novel and innovative therapeutic approach to slow down the progression of the disease caused by genetic defects in the *PRPH2* gene.

This is a very innovative approach that might become a potential treatment in the future for patients suffering from this disease. Especially because ASOs are easy to deliver and well tolerated in the eye. Thus, these findings are very promising to further develop this strategy.

Major comment:

The manuscript show very beautiful data and it is really well written and easy to follow. My main concern is the lack of proper controls for the ASO experiments. The authors have used an ASO previously characterized to downregulate RHO expression. In this context the authors are claiming that by downregulating RHO, the phenotype of mouse models carrying mutations in the *Prph2* gene improve the phenotype. However, it is not discriminated if the delivery of an ASO itself, or the chemical modifications of this ASO can be the inducers of this improvement. For that, members of the Oligonucleotide Therapeutics Society published in open access the guidelines on which controls are needed for the different types of ASOs (<https://www.liebertpub.com/doi/10.1089/nat.2018.0772>). This guidelines have been fully accepted in the field and are very helpful to design experiments. This guidelines were developed after several ASOs showing phenotype improvement that later it was shown to be recapitulated with a non-targeting control ASO. Indicating, that the effect was not directly related to the targeted gene. It is for that reason that the vehicle itself is not enough as a control. Therefore, authors need to perform the experiments with at least one control oligo with similar properties to show that the delivery of any ASO does not improve the phenotype.

Answer: We appreciate the reviewer's suggestion to include a control ASO in our studies, and we have incorporated it in recent experiments over the past several months. P30 *Prph2*^{Y141C/+} mice were left untreated, treated with the vehicle or received control ASO. Functional (ERG for rod and cone responses), histological, and biochemical assessments were conducted 15 days post-treatment, with the data presented in supplementary Figs. 4, 5, 6, respectively. Notably, no differences were observed among the three cohorts in all analyses. The inclusion of these additional data further supports our assertion that downregulation of RHO expression in mutant *Prph2* models improves the phenotype.

Minor comments:

It is not accepted anymore that when referring to humans or disease the word "mutation" is used. Please make sure they are change to variants, pathogenic variants, genetic defects, etc. It is still accepted for mouse, plasmids, etc.

Answer: We have made the appropriate adjustments by replacing the term "mutation" with "pathogenic variant" where it is contextually fitting.

The first time the variants are given, please provide also the c.XXX nomenclature together with the protein.

Answer: We have added the c.422A>G for the Y141C mutation and c.458-460Del for the K153Del mutation. This information is included in the abstract and introduction of the revised manuscript.

In the introduction, the part in which the ASO is mentioned seems to come completely out of the blue. Perhaps some more context would improve readability.

Answer: We have edited this sentence hoping it provides more context as to why we chose to employ this ASO.

Although the ASO is published, please indicate the sequence and chemical modification in the methods.

Answer: The ASO sequence and chemical modifications have been listed in the methods section.

Please indicate the amount of ASO injected in all the figure legends that relate to assessing a parameter upon ASO delivery. This applies to main figures 4, 5, 6, 8; and supplementary figure S6.

Answer: The ASO dosage has been included in the requested figure legends.

Sometimes results are given in mean +/-SD and others mean +/-SEM, can the authors elaborate on the reason for that?

Answer: We have changed all results to represent the mean \pm SD.

REVIEWER COMMENTS

Reviewer #1 (Remarks to the Author):

The authors have improved the manuscript, and I have no additional recommendations.

Reviewer #2 (Remarks to the Author):

The manuscript “Down regulation of rhodopsin is an effective therapeutic strategy in ameliorating peripherin-2-associated inherited retinal disorders” has been resubmitted for revision. As previously indicated, the authors described a potential strategy to ameliorate PRPH2-associated phenotype using antisense oligonucleotides. This is a novel and innovative approach with treatment potential.

The authors have addressed the experimental lack of control oligonucleotide by performing extra experiments and showing that the ASO itself does not cause any difference, and pointing towards a direct effect of rhodopsin downregulation. However, consistency and lack of accuracy is still an issue. Below I listed a list of comments that need to be addressed/revised to ensure the information is correct.

1) According to LOVD, the c.458-460del variant has never been described. Also if it is searched it does not create a deletion of a lysine. In contrast, the p.Lys154del in LOVD is described as c.461_463del, and not what stated in the manuscript. Could the authors verify this information and correct?

2) Also correct nomenclature in humans for pathogenic variants is with the full name (e.g., p.Lys154del), so I would suggest to indicate the full name when referring to human. When referring to the mouse model it can be left in the abbreviation it is now.

3) Also “Del” should not be capitalized (it should read “del”).

4) The term “Pathogenic variant(s)” should only be used in the context of human. The authors have changed this term everywhere including when referring to mouse, which it

also sounds weird because the pathogenic variants were introduced on purpose and therefore they are mutations. Authors should be careful with the nomenclature.

5) ASO sequence description does not match the publication Murray et al. Also it is not mentioned anywhere in the text that this ASO is a GapmeR, which means that it is a hybrid RNA/DNA molecule. The ASO has 2 RNA wings 2MOE/PS with a core of DNA PS. Usually the wings are underlined. This information is missing. Also it is not clear the chemical modifications of the control oligo, which obviously to be a proper control, should be the same as the ASO (namely RNA wings 2MOE/PS with a DNA PS core).

6) How was the control ASO generated? Was selected from previous literature? Was designed by the authors? If the first, please indicate the reference. If the second, please indicate the reasoning in the design.

7) Why the authors decided to only show the control oligo results in the supplementary material? To avoid raising criticism, I would at least suggest to refer in the figure legend to the control results for the same type experiments. Otherwise, at a first sight it seems that the control is missing.

8) Please indicate in all the figure legends the dose of the ASOs. It is not in all of them (especially in the legends using the control ASO).

9) Table 1. The "mRNA ASO1", should be "mRho ASO1"

10) Be consistent with the nomenclature, for example in Figure 6 all figures say P30, P60, P15, etc. but in the legend it says PI-75, which is not in the Figure.

Response to Reviewer 2.

1. According to LOVD, the c.458-460del variant has never been described. Also if it is searched it does not create a deletion of a lysine. In contrast, the p.Lys154del in LOVD is described as c.461_463del, and not what stated in the manuscript. Could the authors verify this information and correct?

In both human and mouse PRPH2 amino acid sequences, two lysines are adjacent at positions 153 and 154, corresponding to c.458-460 and c.461-463 (see below). The human cDNA sequence at this site is **AAG-AAG**. The initial manuscript describing this mutation referred to it as a deletion of 3bp at position 153/154, as it was challenging to discern which AAG was deleted [1].

Following convention, our generated mouse model with a 3 bp deletion at the same site is termed the K153del model. In mice, the amino acid sequence in this region aligns with the human sequence, while the cDNA sequence is **AAA-AAG**. During the generation of the mouse model, we deleted the AAA corresponding to position 153. Despite this, the resulting amino acid sequence mirrors the human condition, presenting a mutant protein lacking one of the two lysines at that site. This mouse line has been previously characterized (see [2]).

Consequently, this mouse line corresponds to the initial patient report in the literature referring to the deletion of 153/154 [3], as well as the information documented in the Leiden Open Variation Database (LOVD) and more recent literature, recognized as c.461_463del, p.Lys154del (db-ID PRPH2_000084) (see [3]).

Human PRPH2

cDNA:421TACCGGGACACAGACACCCCTGGCAGGTGTTTCATG**AAGAAG**ACCATCGACATGCTGCAG480
AA: 141 Y R D T D T P G R C F M **K K** T I D M L Q 160

Mouse Prph2

cDNA:421 TATCGGGATACGGACACCCCGGCCGGTGCTTCATG**AAA****AAG**ACCATCGACATGCTCCAG480
AA: 141 Y R D T D T P G R C F M **K K** T I D M L Q 160

This information has been included in the Methods Section of the revised manuscript. Due to page margin difference, a shorter sequence (5 bp less) is included.

2. Also correct nomenclature in humans for pathogenic variants is with the full name (e.g., p.Lys154del), so I would suggest to indicate the full name when referring to human. When referring to the mouse model it can be left in the abbreviation it is now.

Reference to the human variant has been corrected.

3. Also “Del” should not be capitalized (it should read “del”).

“Del” was changed to “del”.

4. The term “Pathogenic variant(s)” should only be used in the context of human. The authors have changed this term everywhere including when referring to mouse, which it also sounds weird because the pathogenic variants were introduced on purpose and therefore they are mutations. Authors should be careful with the nomenclature.

We eliminated reference to “variant” when describing the mutant mouse

5. ASO sequence description does not match the publication Murray et al. Also it is not mentioned anywhere in the text that this ASO is a GapmeR, which means that it is a hybrid RNA/DNA molecule. The ASO has 2 RNA wings 2MOE/PS with a core of DNA PS. Usually the wings are underlined. This information is missing. Also it is

not clear the chemical modifications of the control oligo, which obviously to be a proper control, should be the same as the ASO (namely RNA wings 2MOE/PS with a DNA PS core).

Description was added in the Methods Section as well as citations for the use of the control ASO.

6. How was the control ASO generated? Was selected from previous literature? Was designed by the authors? If the first, please indicate the reference. If the second, please indicate the reasoning in the design.

New information was inserted in the Methods Section describing the control ASO and the citations.

7. Why the authors decided to only show the control oligo results in the supplementary material? To avoid raising criticism, I would at least suggest to refer in the figure legend to the control results for the same type experiments. Otherwise, at a first sight it seems that the control is missing.

The control data has been moved to the main text.

8. Please indicate in all the figure legends the dose of the ASOs. It is not in all of them (especially in the legends using the control ASO).

Dose was added to those that did not have it already

9. Table 1. The “mRNA ASO1”, should be “mRho ASO1”

Correction has been made in the revised manuscript.

10. Be consistent with the nomenclature, for example in Figure 6 all figures say P30, P60, P15, etc. but in the legend it says PI-75, which is not in the Figure.

Reference to “PI” has been eliminated.

Relevant citations

1. Weleber, R.G.; Carr, R.E.; Murphey, W.H.; Sheffield, V.C.; Stone, E.M. Phenotypic variation including retinitis pigmentosa, pattern dystrophy, and fundus flavimaculatus in a single family with a deletion of codon 153 or 154 of the peripherin/RDS gene. *Arch. Ophthalmol.* **1993**, *111*, 1531-1542.
2. Chakraborty, D.; Conley, S.M.; Zulliger, R.; Naash, M.I. The K153Del PRPH2 mutation differentially impacts photoreceptor structure and function. *Hum. Mol. Genet.* **2016**, *25*, 3500-3514, doi:10.1093/hmg/ddw193.
3. Reeves, M.J.; Goetz, K.E.; Guan, B.; Ullah, E.; Blain, D.; Zein, W.M.; Tumminia, S.J.; Hufnagel, R.B. Genotype-phenotype associations in a large PRPH2-related retinopathy cohort. *Hum. Mutat.* **2020**, *41*, 1528-1539, doi:10.1002/humu.24065.

REVIEWERS' COMMENTS

Reviewer #2 (Remarks to the Author):

The authors have addressed the concerns of this reviewer.

Please check:

- There are still some "Del" instead of "del".
- While the nomenclature of the pathogenic variants have changed in the introduction and discussion, the abstract is not modified.
- Table 1 is corrected only partially, it is still indicated mRNA ASO1.
- Please pay attention to consistency

Response to Reviewer 2.

Please check:

There are still some "Del" instead of "del".

This has been corrected.

While the nomenclature of the pathogenic variants have changed in the introduction and discussion, the abstract is not modified.

The entire abstract has been changed according to Editorial Office requirements.

Table 1 is corrected only partially; it is still indicated mRNA ASO1.

This has been corrected.

Please pay attention to consistency

We have reviewed the manuscript for consistency as suggested.